# Improvement of Microbial Quality, Physicochemical Properties, Fatty Acids Profile, and Shelf Life of Basa (*Pangasius bocourti*) Fillets during Chilling Storage Using Pepsin, Rosemary Oil, and Citric Acid

**DOI:** 10.3390/foods12224170

**Published:** 2023-11-19

**Authors:** Raghda A. Abd El-Fatah, Mahmoud A. Rozan, Hamid M. Ziena, Kálmán Imre, Adriana Morar, Viorel Herman, Heba H.S. Abdel-Naeem

**Affiliations:** 1Department of Food and Dairy Science and Technology, Faculty of Agriculture, Damanhour University, Damanhour 22516, Egypt; raghdaawad41@gmail.com (R.A.A.E.-F.); mahmoud.abdelgalil@agr.dmu.edu.eg (M.A.R.); hamid.ziena@agr.dmu.edu.eg (H.M.Z.); 2Department of Animal Production and Veterinary Public Health, Faculty of Veterinary Medicine, University of Life Sciences “King Mihai I” from Timisoara, 300645 Timisoara, Romania; adrianamo2001@yahoo.com; 3Department of Infectious Diseases and Preventive Medicine, Faculty of Veterinary Medicine, University of Life Sciences “King Mihai I” from Timisoara, 300645 Timisoara, Romania; viorel.herman@fmvt.ro; 4Department of Food Hygiene and Control, Faculty of Veterinary Medicine, Cairo University, Giza 12211, Egypt

**Keywords:** basa fillet, pepsin activity, citric acid, rosemary, microbial quality, deterioration criteria, sensory attributes

## Abstract

Meat discoloration, lipid oxidation, and undesirable texture are inevitable phenomena in basa fish fillets during storage, which in turn limits their exportation as well as decreases consumer acceptability. In addition, increasing consumers’ requirements for high-quality, minimally processed, and ready-to-cook fish fillets with an extended shelf-life is a great challenge, particularly with lifestyle changes. Accordingly, this study aimed to improve the quality, lipid stability, fatty acid profile, and lipid nutritional quality indices (LNQI) of basa fish fillets during chilling storage at 4 °C for 15 days using pepsin enzyme (E, 0.1%), rosemary oil (R, 0.5%), citric acid (CA, 0.5%), and their combination (0.1% E + 0.5% R; 0.1% E + 0.5% CA; and 0.1% E + 0.5% R + 0.5% CA). Our results revealed that all treated samples exhibited a significant increase in protein content, a significant decrease in fat content, and a marked reduction in pH, total volatile base nitrogen (TVBN), thiobarbituric acid (TBA), free fatty acids, and shear force (SF) values in comparison to control ones. Moreover, significant improvements in sensory scores, color stability, fatty acid profile, LNQI, and microbial quality of all treated samples were observed. Such findings were more pronounced in samples treated with a mixture of pepsin, rosemary, and citric acid (TVBN: 2.04 vs. 6.52 mg%; TBA: 0.40 vs. 2.68 mg malonaldehyde/Kg; and SF: 8.58 vs. 19.51 Kgf). Based on the obtained results, there was an extension for the shelf life of all treated basa fish fillet samples, especially in samples treated with a mixture of pepsin, rosemary, and citric acids when compared with the control samples (˃15 days versus 10 days). Additionally, eucalyptol, camphor, isoborneol, and α-pinene are the main components of rosemary, with great antioxidant and antimicrobial activity. In conclusion, the mixture of pepsin, rosemary, and citric acid can be applied easily in the seafood industry and at the household level to provide ready-to-cook fish fillets of high quality with great health benefits.

## 1. Introduction

Fish, mainly *pangasius bocourti* (basa) and *Pangasius hypophthalmus* (tra/swai), are major catfish diversities in recent years, and their production has increased in local and international markets [1]. Vietnam is the largest producer of such catfish since it supplies more than 90% of world exports [2]. There is an enormous potential for the development of convenience fish products from pangasius fish, such as fish fillets, which can be filleted easily owing to the lack of intramuscular pin bones. Typically, it is processed into frozen fillets for domestic consumption, as well as for export to Europe and the USA. Basa has a superior flavor, taste, color, and texture than tra; nonetheless, it has grown slower and cannot sustain aquatic environments compared to tra [3].

The popularity and acceptability of the basa fillet have recently been growing due to its affordable cost, as well as excellent sensory properties, particularly the tender flesh, white meat, sweet taste, delicate flavor, lack of fishy odor and spines, and firm texture after cooking. Likewise, its high nutritional quality involves high protein content, plentiful essential amino acids, and low-fat content, which make such a fillet a favorable choice as a seafood source [4]. On the other hand, a fish fillet is categorized as a perishable commodity with a short shelf life due to its high digestible protein and moisture content. In this regard, deterioration of fish quality caused by enzyme and microbial activity can be reduced by freezing or chilling storage [5]. However, meat discoloration that occurs during storage is an inevitable phenomenon. Accordingly, attempts to maintain color stability contribute significantly to the profit. Furthermore, lipid oxidation of the basa fillet is of great concern to consumers and food industries since it contributes to the development of undesirable sensory quality, besides causing loss of nutritional value.

The changes in consumers’ lifestyles made their demand for high-quality, minimally processed, and ready-to-cook basa fillets with extended shelf-life a great challenge [6]. These challenges have encouraged researchers to develop safe and appropriate techniques for seafood preservation. Moreover, recent interventions in the fish processing sector using natural antimicrobial and antioxidant ingredients based on phytochemicals are gaining significance in improving the safety and quality of fish products, as well as extending their shelf life [7]. Essential oils (EOs) are considered popular natural preservative alternatives to chemical ones. Among these EOs, rosemary (*Rosmarinus officinalis*) exhibits excellent antioxidant and antimicrobial activities owing to its phenolic and flavonoid contents, along with improving the sensory characteristics of food, a matter that warrants its addition to food products [8].

The incorporation of organic acids, such as citric acid, is employed in different fields, such as food, beverages, cleaning/detergent, cosmetics, and pharmaceutical industries [9]. Citric acid is widely used in the food industry as an acidulant, emulsifying, flavoring, and cryoprotectant agent due to its non-toxic and inexpensive additive [9,10]. Moreover, it is incorporated into the seafood marinating process to improve microbial quality, oxidative stability, and sensory characteristics of fish products during storage. The pepsin enzyme is widely used in pharmaceutical, agrochemical, detergent, and leather tanning industries [11], as it is used to produce peptides with high antimicrobial, antioxidant, and antihypertensive activities [12]. It is considered one of the promising protease enzymes that has wide applications in the food industry, as it can serve a beneficial role in food processing. It can be applied in foods to hasten protein hydrolysis and make them easily swallowed without any changes in their original structure, flavor components, and nutrients, even after cooking [13]. In this context, citric acid and pepsin enzymes can be used to enhance the quality of frozen basa fillets and overcome the undesirable texture caused by the freezing process.

The marination process is defined as immersing the meat into a liquid marinade and allowing it to penetrate through diffusion over time. Marinade solutions could be applied by soaking, injecting, or tumbling with aqueous solutions. Immersion, soaking, or marinating is the most practical marinating procedure, which does not require any complicated equipment; therefore, it could be applied easily on the home scale and in the food service sector. However, the main problem of this application procedure is the slower and lower marinade uptake rate, as it needs long marination times [14]. This problem could be overcome by increasing the marinade strength by incorporating tenderizing agents such as enzymes and acids. Despite marination technology being well established in fish industries, no information is yet available about the role of citric acid, pepsin enzyme, and rosemary EOs on the quality of marinated chilled basa fillets. Moreover, to the best of our knowledge, this study is the first concerning the effect of these additives on Lipid Nutritional Quality Indices (LNQI) of the basa fish fillet. In this sense, this study was undertaken to examine the individual effect of citric acid, pepsin enzyme, or rosemary EOs as well as their combination on the sensory attributes, oxidative stability, fatty acid profile, and microbial quality of the marinated basa fish fillet during chilling storage at 4 °C, for 15 days.

## 2. Materials and Methods

### 2.1. Experimental Design

Three independent occasions at separate times (three samples/occasion) were performed to investigate the effect of using citric acid, pepsin enzyme, or rosemary EOs alone and their combination on the sensory attributes, oxidative stability, fatty acid profile, and microbial quality of marinated the basa fish fillet during chilling storage at 4 °C for 15 days.

### 2.2. Basa Fish Fillet Sampling and Preparation of Treatment Solutions

The imported frozen basa fish (*Pangasius bocourti*) fillets (~30 kg) were purchased from a market in the city of Cairo. The samples were transferred in an insulated icebox to the laboratory of Food Hygiene, Faculty of Veterinary Medicine, Cairo University, Egypt (FVMCU), for further processing. Bovine pepsin enzyme powder, citric acid, and rosemary (*Rosmarinus officinalis*) EOs were purchased from Sigma–Aldrich (St Louis, MO, USA). An initial experiment was conducted to choose the best concentration of pepsin, rosemary oil, and citric acid on the quality parameters of the basa fish fillet.

### 2.3. Treatments Application

Treatment conditions for control and all treated groups are shown in Table 1. Firstly, frozen basa fillet samples were thawed in a refrigerator at 4 °C for 12 h. Six groups were prepared by soaking basa fillets in different marinade solutions. The first control (C) group was soaked in distilled water (100 mL DW/Kg fish fillet). The second, third, and fourth groups were treated with a 0.1% pepsin enzyme (E: 0.1 g E + 99.90 mL DW/Kg fish fillet), 0.5% water soluble rosemary EOs (R: 0.5 mL R + 99.50 mL DW/Kg fish fillet), and 0.5% citric acid (CA: 0.5 g CA + 99.50 mL DW/Kg fish fillet), respectively. Meanwhile, the fifth group was treated with a 0.1% pepsin enzyme + 0.5% rosemary EOs (E + R: 0.1 g E + 0.5 mL R + 99.40 mL DW/Kg fish fillet), the sixth group was treated with a 0.1% pepsin enzyme + 0.5% citric acid (E + CA: 0.1 g E + 0.5 g CA+ 99.40 mL DW/Kg fish fillet), and the seventh group was treated with a 0.1% pepsin enzyme + 0.5% rosemary EOs + 0.5% citric acid (E +R +CA: 0.1 g E + 0.5 mL R + 0.5 g CA+ 98.90 mL DW/Kg fish fillet). Treated and control groups were packed in sterile polyethylene bags with their solutions and stored in a refrigerator at 4 °C for 15 days. All samples were examined on the first day of treatment application and every 3 days for up to 15 days of chilling storage.

### 2.4. Investigations

#### 2.4.1. Measurement of Pepsin Enzyme Activity

Pepsin activity was measured using the method described by Brodkorb et al. [15] based on Minekus et al. [16]. A total of 100 µL of pepsin enzyme was added into the Eppendorf tube containing 500 µL of hemoglobin (2%) in the buffer at a pH of 2. The mixture was mixed and incubated at 37 °C for 10 min. One milliliter of 5% trichloroacetic acid (TCA) was added to stop the reaction, then shaken vigorously, and the mixture was centrifuged at 6000× *g* for 30 min. The absorbance of the supernatant was measured at a wavelength of 280 nm using a spectrophotometer (Unico 1200, Dayton, NJ, USA) against the blank prepared by adding pepsin enzyme after TCA. The activity of pepsin was calculated using the following Equation (1).
(1)$$U=\frac{(As-Ab)\times 1000}{\left(T\times VE\right)}$$U: activity units (AU/mL). As: absorbance of the sample. Ab: absorbance of the blank. T: incubation time. VE: enzyme volume.

#### 2.4.2. Analysis of Rosemary Essential Oil

The chemical compounds of rosemary essential oil were analyzed using gas chromatography–mass spectroscopy (Agilent 8890-5977B, GC/MSD, Santa Clara, CA, USA) equipped with a HP-5MS capillary column (30 m × 0.25 mm × 0.25 mm) according to the method described by Abd El-Wahab et al. [17]. The oven temperature was maintained initially at 50 °C, followed by an increase to 220 °C at a rate of 5 °C min^−1^, and lastly, reaching 280 °C at a rate of 20 °C min^−1^, with a hold period of 5 min. Helium was used as the carrier gas with a flow rate of 1 mL min^−1^. Thirty microliters from the essential oil were dissolved in 1 mL of diethyl ether, and then 1 µL from this solution was injected into the GC at an injector temperature of 230 °C and a split ratio of 1:50. Mass spectra were obtained at 70 eV via electron ionization (EI) and the mass range was from 39 to 500 amu. The isolated peaks were identified by matching them with data from the library of mass spectra (National Institute of Standards and Technology, NIST).

#### 2.4.3. Examination of Basa Fish Fillets

##### Proximate Compositional Analysis

Moisture, protein, fat, and ash contents (g/100 g) of basa fish fillet samples were analyzed according to the official method of AOAC with procedure numbers AOAC 925.45, AOAC 981.10, AOAC 991.36, and AOAC 923.03, respectively [18]. Moisture contents were examined by drying 3 g of a sample at 100 °C until a constant weight was attained. However, protein content was examined according to the Kjeldahl technique, and nitrogen content was converted into crude protein using a factor of 6.25. Fat content was analyzed using the Soxhlet apparatus, and ash content was analyzed by ignition of the samples in a muffle furnace for 5 h at 500 °C.

##### Measurement of pH

The pH value was measured using a digital pH meter (Lovibond, Senso Direct) according to the technique outlined by Lee and Shin [19] after calibration of the pH meter with two buffer solutions (4.0 and 7.0) and blending 5 g of basa fish fillet samples with 20 mL distilled water.

##### Total Volatile Base Nitrogen (TVBN)

Total volatile base nitrogen (TVBN) was measured according to the method proposed by Malle and Tao [20] using a steam distillation technique. For the preparation of fish extracts, 100 g of basa fish fillet samples were homogenized with a 200 mL aqueous solution of 7.5% trichloroacetic acid (TCA, *v*/*v*). The homogenate was centrifuged at 3000 rpm for 5 min; then, the supernatant liquid was filtered through the Whatman No. 1 filter paper. A total of 25 mL of fish extract and 5 mL of 10% NaOH (*w*/*v*) aqueous solution were mixed in a Kjeldahl distillation tube, and steam distillation was performed. A total of 49 mL of the distillate was obtained in a receiving flask containing boric acid (10 mL, 4%, *v*/*v*) aqueous solution and 0.04 mL of methyl red and bromocresol green indicator. The distilled TVBN alkalinized the boric acid solution and changed its color from pink to green. Ultimately, the distillate was titrated with 0.1 N of sulfuric acid solution until the color returned to pink and complete neutralization occurred. TVBN values were determined from the volume of titrated sulfuric acid (0.1 N) multiplied by 16.8, and the result was expressed in mg nitrogen/100 g of sample.

##### Fat Oxidation Parameters

Thiobarbituric acid (TBA) was examined according to the procedure recommended by Tarladgis et al. [21]. Ten grams of basa fish fillet samples were homogenized with 97.5 mL of distilled water and 2.5 mL of HCl solution (4 N) for 2 min. The homogenate was placed in a distillation tube, and steam distillation was completed till 50 mL of distillate was collected. Five milliliters from each the distillate and TBA reagent (0.02 M) in acetic acid (90%) were added into a test tube and heated in a boiling water bath for 35 min to obtain the TBA-MDA complex followed by cooling for 10 min under tap water. The absorbance was measured at 538 nm using a spectrophotometer (Unico 1200, USA) against a blank containing 5 mL of each TBA reagent and distilled water. Based on the standard calibration curve, the malonaldehyde (MDA) amount was calculated, and a conversion factor 7.8 was adopted to convert the readings of TBA-MDA to the TBA value (mg MDA/kg).

Determination of the content of free fatty acids (FFAs) and acid number (AN) in basa fish fillet samples was performed according to the technique outlined by AOCS [22]. Fifteen grams of the basa fish fillet sample were blended in 60 mL of each methanol and chloroform solvent. After 24 h, 48 mL of distilled water was added, and oil was gathered. The extracted oil was titrated by sodium hydroxide (0.1 N) in the presence of phenolphthalein (1%). FFAs (% as Oleic acid) and AN (mg NaOH/g) were calculated using the following Equations (2) and (3):(2)$$\mathrm{FFAs}\;(\%)=\frac{mL\;of\;alkalix\;Nx28.2}{W}$$N: Normality of NaOH solution. W: Weight of oil (g).
(3)$$Acid\;number\;(mg\;NaOH/g)=1.99\times FFAs\left(\%\right)$$

##### Color Evaluation

The color of basa fish fillet samples was measured according to the method recommended by Shin et al. [23] using a Chroma Meter (Konica Minolta, model CR 410, Tokyo Japan) calibrated with a white plate and light trap supplied by the manufacturer and programmed to use a D65 Commission Internationale de l’Eclairage (CIE) standard illuminant, 10° observer with an illuminating/viewing aperture size of 11 mm, and a bloom time of 30 min. The average value of three readings was recorded for each sample and expressed as CIE lightness (L*), redness (a*), and yellowness (b*). The whiteness index (WI) was calculated using the following Equation (4).
WI = 100 − {(100 − L*)^2^ + a*^2^ + b*^2^}^1/2^(4)

##### Measurement of Shear Force

The shear force value of basa fish fillet samples was measured using a Warner–Bratzler shear force device attached to an Instron Universal Testing Machine (Model 2519 105; Instron Corp., Canton, MA, USA) as described by Shackelford et al. [24].

##### Measurement of Cooking Loss

All basa fish fillet samples were cooled down to room temperature after cooking (in an oven at a core temperature of 75 °C for 30 min), and the cooking loss percentage was calculated from the following Equation (5) after weighing the raw (Wr) and cooked (Wc) samples.
(5)$$\mathrm{Cooking}\;\mathrm{loss}\;\%=\frac{Wr-Wc}{Wr}\times 100$$

##### Fatty Acids Profile Analysis

The fatty acid composition was determined through the transmethylation of the fatty chains to fatty acid methyl esters (FAMEs) according to the method recommended by ISO-12966–4 [25] with some modifications. The separation of FAMEs was performed using gas chromatography (HP 6890 plus, Hewlett Packard, Palo Alto, CA, USA) equipped with a capillary column (Supelco™ SP-2380, Sigma-Aldrich, USA) with dimensions of 60 m × 0.25 mm × 0.20 μm and a FID Detector. The column temperature was 140 °C with a holding time of 5 min followed by an increase to 240 °C at a rate of 4 °C min^−1^ and held at this temperature for 10 min. The helium was used as a carrier gas with a flow rate of 1.2 mL min^−1^. One microliter of the sample volume (in n-hexane) was injected at a temperature of 250 °C and a splitting ratio of 100:20. FAMEs were identified by matching their relative and absolute retention times with authentic standards of FAMEs (SupelcoTM 37 component FAME mix). The fatty acid composition was reported as a relative percentage of the total peak area.

Lipid Nutritional Quality Indices (LNQI) were assessed from the data on fatty acids. The atherogenic index (AI) and thrombogenic index (TI) were calculated from Equations (6) and (7), respectively [26], while the nutritive value index (NVI) was calculated from Equation (8) [27].
(6)$$\mathrm{AI}=\frac{\left(C12:0+\left(4\times C14:0\right)+C16:0\right)}{\left(\Sigma MUFA+\Sigma n-6+\Sigma n-3\right)}$$
(7)$$TI=\frac{C14:0+C16:0+C18:0}{\left[(0.5\times \Sigma MUFA)+(0.5\times \Sigma n-6)+(3\times \Sigma n-3)+(\Sigma n-3/\Sigma n-6)\right]}$$
(8)$$NVI=\frac{C18:0+C18:1}{C16:0}$$

##### Bacteriological Examination

For the enumeration of aerobic plate count (APC) and the psychrotrophic bacteria, double sets of inoculated Standard Plate Count Agar plates (Oxoid CM 463) were incubated at 32 °C, for 48 h [28], and at 7 °C, for 7 days [29], respectively. However, Enterobacteriaceae bacterial counts were enumerated by incubating the inoculated Violet Red Bile Glucose agar (Oxoid CM 1082) at 37 °C for 24 h [30].

##### Sensory Examination

Sensory analysis of basa fish fillet samples was performed by trained fifteen panelists. Such panelists were selected according to their experience in the sensory assessment of fish and fish products. Furthermore, they attained a preparatory session connected to a descriptive analysis of the basa fish fillet, according to ISO-11035 [31]. Retraining was also carried out continuously during the assessment period of 15 days. Three basa fish fillets from each group were cooked in an oven (D-63450, Heraeus, Hanau, Germany) till the core temperature reached 75 °C for 30 min. The samples were served in random order, and the sensory characteristics were assessed using the nine-point hedonic scale where 9 refers to extremely like and 1 refers to extremely unlike for the following characteristics: appearance, color, flavor, tenderness, juiciness, and overall acceptability. The evaluations were performed in distinct booths with controlled temperatures, adequate lighting, and free from odor, and the panelists used water to clean their palate between each sample.

### 2.5. Statistical Analyses

All measurements were carried out in triplicates, and all data were analyzed using SPSS statistics 27.0 for Windows, expressed as mean ± SE, and compared using a one-way analysis of variance (ANOVA). The significance was determined using the least square difference test (LSD) procedure, and the main effects were considered significant at the *p* < 0.05 level.

## 3. Results and Discussion

### 3.1. Pepsin Enzyme Activity

The pepsin enzyme can be assayed by many characteristics, such as enzyme activity, enzyme concentration, and specific activity. The specific activity of an enzyme implies the amount of enzyme activity per unit mass of protein, which is an important indicator of enzyme homogeneity and quality [32]. Our results revealed that the activity value of the pepsin enzyme used in the present study was 223.34 U/g with a protein concentration of 0.0361 mg/g; accordingly, its specific activity was 6186.7 U/mg (Table 2). Those characteristics confirm the tremendous potency of the pepsin enzyme.

Lower specific activity (1805.17 U/mg) of the pepsin enzyme extracted from yellowfin tuna stomach with a higher activity value (523.5 U/mL) and protein concentration (0.29 mg/mL) than the obtained results were observed by Nurhayati et al. [33]. Moreover, Pasaribu et al. [34] recorded that the specific activity of the pepsin enzyme extracted from the visceral wall fluid of tuna was 4.274 U/mg. Zhao et al. [32] reported that the specific activity of the pepsin enzyme improved by increasing the purification factor. Additionally, Bax et al. [35] noticed that heating can accelerate the degradation potential of meat by the pepsin enzyme, subsequently improving pepsin enzyme activity but at a certain temperature limit as well as with a longer digestion time. They explained such effects by increasing the exposure potential of the pepsin enzyme to cleavage sites with heating. In this circumstance, pepsin activity seems to be temperature-dependent. In another study, Kudryashov et al. [36] found that microencapsulation of the pepsin enzyme is an effective approach to improving the quality of meat products by maintaining high enzyme activity for a long time.

### 3.2. The Chemical Compounds of Rosemary Essential Oil

It is noteworthy that rosemary essential oil contains many volatile organic compounds with great health benefits and technological importance. Sixteen volatile compounds representing 99.98% of the total oil were identified, and the relative percentage of each component varied markedly (Table 3). The main components of rosemary EO used in the present study were eucalyptol (33.15%), camphor (18.84%), isoborneol (13.53%), and α-pinene (10.42%). Besides the aforementioned compounds, there were several compounds detected with a considerable level, such as α-terpineol (5.61%), camphene (4.89), p-cymene (3.28), D-limonene (2.59), bornyl acetate (1.67%), and terpinen-4-ol (1.41%) (Table 3).

Our findings are comparable with those of Diniz-Silva et al. [37] and Dhouibi et al. [38], who found that the most abundant compounds in rosemary EOs were eucalyptol, camphor, and α-pinene. In the same regard, the most predominant compounds in rosemary EOs originating from Russia and Serbia were α-pinene, eucalyptol, and camphor [39]. Conversely, the principal compounds of rosemary EOs were limonene and then camphor [40]; however, in another study, they were piperitone, followed by linalool and α-pinene [41]. The diversity in the chemical composition of rosemary EOs, which in turn influences their biological activity, may be due to intrinsic factors such as plant age and genetics as well as extrinsic factors such as extraction methods, climate, and cultivation conditions [42].

### 3.3. The Physicochemical Characteristics of Basa Fish Fillets

#### 3.3.1. Proximate Compositional Analysis of Basa Fish Fillets

The results obtained in the present study revealed that there was a significant (*p* < 0.05) increase in moisture content and a significant (*p* < 0.05) decrease in ash content in all treated groups, except rosemary-treated samples, when compared with their counterpart control samples. In addition, all treated groups induced a significant (*p* < 0.05) increase in protein content, along with a significant (*p* < 0.05) decrease in fat content (Table 4).

Our observation with respect to the high moisture content in the control basa fillet was in agreement with that previously reported in *Pangasius bocourti* [43] and *Pangasius hypophthalmus* [44,45]. The possible reason behind this finding is due to the addition of water-binding additives with high freezing capacity, such as polyphosphate, during the processing of basa fillets to increase water retention by proteins and reduce the thaw drip [43]. Conversely, Deepitha [46] and Mostafa et al. [47] reported lower moisture and higher protein content in fresh pangasius fillets than the obtained results. In another study, Chakma et al. [48] observed a significantly higher moisture, fat, and ash content with a significantly lower protein content in wild pangasius than in farmed pangasius, and they explained that due to differences in food availability, environmental conditions, and habitat.

The addition of pepsin, rosemary oil, and citric acid, in the present study, significantly increases the protein and moisture content of the treated fish fillet owing to the proteolytic effect of pepsin (Table 2) and their low pH values (Table 5), subsequently increasing protein dissolution and decreasing water-binding ability. Our findings regarding the effect of citric acid on the chemical composition of basa fish fillets are in harmony with those demonstrated by Klinmalai et al. [10], who observed a significant increase in protein content of frozen fish fillets treated with 2% and 5% citric acid. Similarly, Emir Çoban and Özpolat [49] found that rosemary extract had no substantial effect on the moisture content of fish fillets. In contrast to our finding, Nurhayati et al. [33] indicated that treatment of red tilapia surimi with the pepsin enzyme extracted from tuna stomachs has non-significant effects on its chemical composition.

#### 3.3.2. pH and Total Volatile Base Nitrogen (TVBN) of Basa Fish Fillets

Total volatile base nitrogen is a group of biogenic amines, including trimethylamine, dimethylamine, and ammonia. TVBN is considered a tremendous index used to assess the quality and shelf life of seafood products. The results reported herein, in the present study, revealed that treatment of basa fish fillets with pepsin, rosemary, citric acid, or their mixture resulted in a significant (*p* < 0.05) reduction in pH and TVBN values in comparison to control groups throughout the storage period. Such a finding was more pronounced in samples treated with a mixture of pepsin, rosemary, and citric acid (Table 5).

Our data showed that the pH value in control untreated basa fillets at zero time of examination was 6.40 and started to increase progressively till reaching 7 on day 15 of chilling storage.

A high pH value, which is close to 7, was reported by Leksono et al. [50] in fresh pangasius fillets. In another study, a wide range of pH (5.88–7.93) values in *Pangasius hypophthalmus* fillets was recorded by Guimarães et al. [45] as this parameter is strongly affected by many factors, for instance, catching, processing, storage, and the type of additives used during the processing such as polyphosphate. This observation was also confirmed by Guimarães et al. [45], who found that the increase in polyphosphate to a concentration of more than 2 g/100 g could increase the pH value of the pangasius fillet to more than 7.

Regarding the decrease in the pH value in citric acid-treated samples, similar results were reported by Klinmalai et al. [10], who noticed that treatment of frozen fish fillets with citric acid resulted in a significant decrease in pH (4.51–5.56) values. Moreover, the pH value of the tilapia surimi was significantly decreased with increasing pepsin concentration, which may be due to the low pH (2.5) value of pepsin [33]. Likewise, a decrease in the pH value of the rosemary-treated sample in the present study was elucidated by the presence of phenolic compounds in rosemary EOs, which retard the growth of spoilage bacteria and subsequently decrease the production of alkaline compounds. However, no significant change in pH values of rosemary-treated fish fillets was observed by Gao et al. [51] and Karoui and Hassoun [52].

Our findings were comparable with those of Karoui and Hassoun [52], who obtained a significant decrease in TVBN values of rosemary-treated mackerel fillets from day 6 till the end of chilling storage (15 days) and they attributed that to the antibacterial activity of the phenolic compounds in rosemary EOs. Furthermore, rosemary extract combined with nisin significantly reduced the TVBN content of fish fillets during the chilling period as well as improved their shelf life [51]. In addition, the treatment of fish fillets with a chitosan and citric acid combination induced a significant decrease in TVBN starting from day 8 till the end of chilling storage (12 days) when compared with the control sample [53]. On the other hand, Khalafalla et al. [54] recorded a non-significant decrease in TVBN in rosemary-treated Nile tilapia fillets, compared to the control one, which indicates that rosemary extract has poor action against enzymatic and microbial activities.

Our result with respect to the significant decrease in TVBN of citric acid and pepsin-treated samples may be due to their low pH value, which retard the growth of spoilage bacteria and accumulation of alkaline products such as ammonia. However, the significant decrease in TVBN in rosemary-treated samples may be attributed to the presence of essential compounds that possess antimicrobial and antioxidant activities (Table 3). Similarly, Senanayake [55] found that eucalyptol, camphor, α-pinene, and bornyl acetate are the major compounds responsible for the antimicrobial activity of rosemary, a matter that supports our findings (Table 3).

#### 3.3.3. Fat Oxidation Parameters of Basa Fish Fillets

To obtain a comprehensive judgment about the degree of fat oxidation in basa fish fillets-treated with pepsin, rosemary, citric acid, and their combination fat oxidation parameters comprising TBA, FFAs, and AN are examined. TBA is considered an appropriate indicator of the quality of seafood products as it refers to the degree of lipid oxidation inside the fish’s flesh. Likewise, the breakdown of phospholipids and triglycerides resulted in the production of FFAs, which undergo further oxidation and produce compounds responsible for the undesirable flavor and taste of fish and fish products. In this respect, FFAs can be used to assess the degree of lipolysis in fish flesh.

Unlike mammalian meat, the foremost contributor to lipid oxidation in fish meat is hemoglobin, owing to the inappropriate bleeding techniques during processing. Accordingly, iron or haem is released from the fish protein, promoting lipid oxidation in fish meat. It is conspicuous from the obtained results that there were significant (*p* < 0.01) reductions in TBA, FFAs, and AN of all treated groups during the entire storage period, especially in those treated with a mixture of pepsin, rosemary, and citric acid when compared with control samples (Table 5). The obtained low TBA value in the citric acid-treated sample may be due to its ability to chelate prooxidant metals, forming a thermodynamically stable complex, consequently lowering their redox potentials. However, improving the lipid stability in rosemary-treated samples is due to their phenolic content, which possesses antioxidant activity. In addition, the significant decrease in the TBA value of pepsin-treated samples is linked to its antimicrobial activity arising from its low pH, which could reduce or eliminate microbial rancidity. Such clarification is also reported by Mansour et al. [56].

The results obtained are in accordance with those demonstrated by Gao et al. [51], who noticed that rosemary extract significantly decreased TBA values and strongly inhibited lipid oxidation of fish fillets during chilling storage. Furthermore, the TBA value decreased significantly in frozen fish fillets soaked with citric acid [10]. A good synergistic effect was obtained to reduce lipid oxidation when combined nisin with rosemary extract or citric acid [57], which supports our findings. Moreover, pangasius fish treated with seaweed extract [46], as well as spice extract [58], showed low TBA values, as compared with the control sample owing to their phenolic content, which delayed lipid oxidation. A similar trend of the FFA values was obtained by Rathod and Pagarkar [59] during the chilling storage of pangasius fish. However, seaweed extracts dipped pangasius fillets were stable with low FFA values till the end of chilling storage for up to 20 days due to its polyphenols content [46].

#### 3.3.4. Color, Shear Force, and Cooking Loss % of Basa Fish Fillets

Many attempts have been conducted in the present study to devise objective physical and chemical techniques for evaluating eating quality traits in contrast to the subjective assessments of the panelists. Therefore, the degree of color variation was measured using the instrumental color evaluation. It is noteworthy that meat color strongly affects consumers’ decisions to purchase, which refers to its quality and freshness. Data of color assessment clearly revealed that, among all treated samples, using citric acid either alone or in combination with pepsin or with rosemary and pepsin induced significant (*p* < 0.01) increases in lightness (L*) and redness (a*) values. However, there was a significant (*p* < 0.01) reduction in yellowness (b*) values in all treated samples when compared with their counterpart control samples (Table 6).

Whiteness is an important parameter for estimating fish quality, which can be enhanced by removing blood, myoglobin, and lipids. Our findings in this study were supported by Klinmalai et al. [10] and Gu et al. [60], who observed that citric acid significantly increased the whiteness of fish fillets and fish products, respectively, to meet the consumer preference for white fish. Moreover, rosemary EOs could maintain the color of fish fillets due to the antioxidant activity of these EOs, which retard lipid oxidation and metmyoglobin formation [52]. Likewise, Tironi et al. [61] observed that the reduction in red coloration of sea salmon was significantly decreased with increasing rosemary extract concentration. In this perspective, a positive correlation between TBA and b* values of the fish was obtained by Karoui and Hassoun [52]. In contrast, Nurhayati et al. [33] observed a decreasing trend of whiteness as the concentration of pepsin increased in the surimi and attributed that to the brownish color of used tuna stomach pepsin.

Meat tenderness is closely linked to the shear force value, which influences the meat quality and consumer preference for this meat [62,63]. In the present study, shear force values were significantly (*p* < 0.05) decreased in all treated samples as compared to control ones (Table 6). Such an observation may be attributed to their low pH values, which causes protein denaturation and weakening of connective tissue. These results were in agreement with those reported by Gu et al. [60], who found that citric acid could significantly enhance the textural properties of restructured fish and surimi products. In the same regard, there was a significant improvement in the texture properties of fish fillets treated with rosemary EOs [52] and a combination of rosemary EOs with nisin [51].

The impact of pepsin enzyme, rosemary EOs, and citric acid alone and their combination on the cooking loss percentage of basa fillets during chilling storage for 15 days is presented in Figure 1.

Cooking loss % was significantly (*p* < 0.01) increased in all treated groups, except rosemary-treated samples, during the entire storage, compared to the control samples. Such an observation was explained by their low pH values in addition to the proteolytic activity of the pepsin enzyme, which resulted in protein denaturation with a subsequent decrease in its water-binding capacity. Gu et al. [60] reported that citric acid significantly decreased the cooking loss % of restructured fish products; however, it remarkably increased the cooking loss of surimi products. The authors attributed that to the tight structure of citric acid-treated restructured fish products, which increases the water-holding capacity, nonetheless, increases protein denaturation and decreases water encapsulation accompanied by the decrease in the water-holding capacity of citric acid-treated surimi. In addition, Klinmalai et al. [10] recorded an increasing trend in cooking loss % as the concentration of citric acid increased in frozen fish fillets.

### 3.4. Fatty Acids Profile of Basa Fish Fillets

A total of 19 fatty acids were identified in the basa fish fillet, and the most abundant fatty acid was palmitic acid (C16:0), with a mean value of 31.47%. The results obtained in the present study revealed that the total saturated fatty acids (SFAs), monounsaturated fatty acids (MUFAs), polyunsaturated fatty acids (PUFAs), and total unsaturated fatty acids (UFAs) content of the basa fish fillet were 64.07, 27.70, 8.23, and 35.93%, respectively (Table 7). Similar to our finding, Ho and Paul [64] found that the most abundant fatty acids were SFAs, followed by MUFAs and PUFAs in the pangasius fillets from Vietnam. Conversely, Chakma et al. [48] reported that the most predominant fatty acids were MUFAs, followed by SFAs in both wild and farmed pangasius; however, PUFAs were detected only in wild pangasius. Muhamad and Mohamad [65] noticed that PUFA content was lower in freshwater fish than the corresponding levels in marine fish due to freshwater fish feeding mainly on plant materials; nevertheless, marine fish feed more on zooplankton that is rich in PUFA. In this context, the fatty acid composition of the pangasius fillet is variable as it is influenced by several factors, such as age, sex, season, diet, water temperature, degree of salinity, and the extraction method. Such variation could affect the organoleptic properties and nutritional value of fish meat.

The main SFAs of the basa fish fillets examined in this study are palmitic acid (31.47%), myristic acid (12.97), and stearic acid (11.82%). In addition, the most dominant MUFA is oleic acid (26.18), which belongs to n-9 fatty acids, while among PUFAs, linoleic acid (7.08%) is the most abundant (Table 7). Such observations are in agreement with those reported by Sokamte et al. [66], except for myristic acid, which was detected in very small amounts (0.25%). Furthermore, treatment of basa fish fillets with f pepsin, rosemary, citric acid, and their combination resulted in a significant (*p* < 0.05) increase in MUFAs, PUFAs, UFAs, and PUFAs/SFAs, accompanied by a significant (*p* < 0.05) decrease in SFAs compared to control samples (Table 7). These findings are explained by the antioxidant activity of rosemary EOs and citric acid, a matter which substantiates our results of fat oxidation parameters (Table 5). Surprisingly, all basa fillet-treated samples contain bioactive conjugated linoleic acid (CLA) in contrast to the control samples. Consumption of CLA has a marked health-promoting effect due to its anti-diabetic, anti-cancer, anti-obesity, anti-atherosclerosis, and antioxidant activity [67].

Lipid Nutritional Quality Indices, including PUFAs/SFAs, n-6/n-3, AI, TI, and NVI, are essential to be applied in order to evaluate the health benefits as well as nutritional aspects of lipid fractions. The PUFAs/SFAs ratio is used to assess the dietary quality index of the lipid fraction. In this circumstance, the PUFAs/SFAs ratio should be ≥ 0.4 for healthier diets [68]. This recommended ratio was provided only in samples treated with mixtures of pepsin–citric acid (0.41) and pepsin–rosemary–citric acid (0.45) (Table 7). A comparable lower PUFAs/SFAs ratio than the recommended level was recorded by Chakma et al. [48] in both wild (0.2) and farmed pangasius (0.26).

The n-6/n-3 ratio is also a fundamental index used to evaluate lipid quality and assist in the treatment and prevention of numerous diseases. However, eating habits have changed over the years, and Western diets have become rich in n-6 PUFAs and poor in n-3 PUFAs, producing an unhealthy ratio of n-6:n-3 (17 to 20:1), which appears to be much higher than the healthy ratio (2.5 to 8:1) recommended by the WHO/FAO [69]. Furthermore, to prevent the risks of obesity, cardiovascular diseases, and chronic diseases, a ratio of 10:1 for n-6/n-3 was recommended by Simopoulos [70]. It is noticed from our results that all basa fish fillet-treated samples obtained the n-6/n-3 ratio within the recommended limit (Table 7). A higher n-6/n-3 ratio (9.51–11.33) of *Pangasius bocourti* fillets was noticed by Thammapat et al. [71] than the obtained result in control untreated samples (10.76).

The atherogenic and thrombogenic indices are very useful tools that indicate the chance for promoting platelet aggregation and assessing the risk of food to develop heart diseases. Likewise, such indices give an indication of the nutritional quality of lipids since lower values for these indices indicate healthier food with good nutritional quality of fatty acids with subsequent greater probability for the prevention of heart diseases [72]. Our findings revealed that treatment of the basa fish fillet with pepsin, rosemary, citric acid, or their combination induced a significant (*p* < 0.05) decrease in AI and TI along with a significant (*p* < 0.05) increase in NVI (Table 7), a matter which confirms that they provide great health benefits for consumers. Chakma et al. [48] observed that wild pangasius had low AI and TI values when compared with the farmed pangasius.

### 3.5. Microbial Quality of Basa Fish Fillets

The aerobic plate counts (APC), psychrotrophs, and Enterobacteriaceae counts of all basa fillets treated groups were significantly (*p* < 0.05) lower than those of the control samples, at the day 0 of chilling storage, and thereafter until the end of the storage period (15 days). Such observation was more noticeable in samples treated with a mixture of pepsin–rosemary–citric acid. Interestingly, APC count was under detectable level (<2 Log_10_ CFU/g) in samples treated with each of pepsin, rosemary, citric acid, and pepsin–rosemary mixture during the first 3 days of storage, as well as in samples treated with mixtures of pepsin–citric acid and pepsin–rosemary–citric acid during the first 9 days of storage. However, psychrotrophs and *Enterobacteriaceae* counts were under detectable level (<2 Log_10_ CFU/g) in samples treated with pepsin, rosemary, or citric acid during the first 6 days of storage and in samples treated with mixtures of pepsin–rosemary, pepsin–citric acid, and pepsin–rosemary–citric acid during the first 9 days of chilling storage (Table 8).

These results obviously demonstrated that the rosemary EOs, citric acid, and pepsin enzyme had a strong inhibitory effect on microbial growth during chilling storage of basa fish fillet. This observation is in accordance with those published by Gao et al. [51], who obtained a significant reduction in APC of rosemary-treated fish fillet samples starting on the 9th day of chilling storage as compared with control ones. Additionally, dipping the fish fillet in a brine containing ascorbic acid (0.1%) and citric acid (1.0%) induced a significant reduction in APC and psychrotrophic count starting from the 9th and 6th days of chilling storage, respectively, compared to control samples [73].

In another study, pangasius fillets treated with seaweed extracts exerted a significant reduction in APC from the beginning of the 4th day till the end of chilling storage for up to 20 days [46]. Improving the microbial quality of basa fish fillets, in the present study, using rosemary EOs, citric acid, or pepsin enzyme may be due to their low pH value. Likewise, Brul and Coote [74] attributed the antimicrobial activity of citric acid to the ability of the uncharged form of this acid to penetrate through the microbial cell membrane and acidify its cytoplasm. Furthermore, rosemary EOs are rich in phenolic compounds, which possess antibacterial activity. It is noteworthy that our results of microbial quality of basa fish fillets-treated samples follow the same pattern as their results of protein deterioration and fat oxidation parameters (Table 5).

### 3.6. Sensory Characteristics of Basa Fish Fillets

The changes in sensory scores of treated and control untreated basa fish fillet samples during the chilling storage are depicted in Figure 2, Figure 3 and Figure 4. The results revealed that all treated samples exhibited significant (*p* < 0.05) improvement in appearance, color, flavor, tenderness, and overall-acceptability scores during chilling storage compared to control samples. Nonetheless, juiciness scores were significantly (*p* < 0.05) decreased in all treated groups except rosemary-treated samples during the entire chilling storage compared to control ones, which could be due to the increase in their cooking loss % (Figure 1). It is also clear that the results of sensory scores substantiated the data of shear force and color values (Table 6) as well as cooking loss % (Figure 1).

Our findings were consistent with those reported by Elhafez et al. [75], who observed significant improvement in sensory scores, including color, texture, and odor of Nile tilapia fish fillet treated with 1.5% rosemary oil. The high color and flavor scores in rosemary-treated samples obtained in this study could be correlated to the antioxidant components that delay pigment and lipid oxidation, as well as the volatile oils that are responsible for flavor (Table 3). Such an observation was also reported by Senanayake [55]. Similarly, Gao et al. [51] found that combined rosemary extract with nisin could delay odor and color deterioration in fish fillets as well as extend their shelf life during chilling storage owing to their antioxidant activity against lipids and pigments. Moreover, citric acid is commonly used as a flavoring agent and preservative in food [76], a matter which supports the obtained results. In addition, the pepsin enzyme induced significant improvement in the sensory scores of red tilapia surimi [33].

### 3.7. Shelf life of Basa Fish Fillets

It is noteworthy that the APC of control samples exceeds the acceptable limit (not more than 6 Log_10_ CFU/g) described by ES-3494 [77] starting from 11 days of chilling storage and within the acceptable limit for all treated samples during the storage period. Despite TVBN and TBA values of control, all treated samples were within the acceptable limit (not more than 30 mg% for TVBN and 4.5 mg malonaldehyde/Kg) described by ES-3494 [77]. In this sense, based on the microbial and sensory data, there was an extension for the shelf life of all treated basa fish fillet samples, especially in samples treated with a mixture of pepsin, rosemary, and citric acids when compared with the control samples (˃15 days versus 10 days). Accordingly, the results highlighted that treatment of basa fish fillets with rosemary EOs, citric acid, pepsin enzyme, or their combination could improve microbial quality, lipid stability, fatty acid profile, and sensory characteristics, accompanied by an extension in their shelf life and such effects were pronounced in pepsin–rosemary–citric acid- treated samples. Based on the sensory examination, Jeyakumari et al. [58] and Jeyakumari et al. [78] obtained an extension in the shelf life of spice extract-treated pangasius chunks (9 versus 15) and chitosan-treated pangasius products (10 versus 17) compared to control samples, respectively. Despite the nearly similar shelf life obtained in the present study for the control untreated basa fillet, a higher shelf life for all treated groups was recorded than in the aforementioned studies.

## 4. Conclusions

In recapitulation, the major quality issues associated with the storage of basa fish fillets are lipid oxidation, short shelf life, and color deterioration with subsequent loss of their characteristics whitish color. Based on the obtained promising results, the expected shelf life of control chilled basa fish fillet samples was 10 days, and after that, they started to deteriorate till the end of chilling storage (15 days). On the other hand, all treated basa fish fillet samples, especially those treated with the mixtures of pepsin enzyme, rosemary, and citric acids, could maintain their sensorial attributes along with their low microbial load without any obvious deteriorative changes by the end of 15 days. Accordingly, using this mixture is considered a good choice to overcome all aforementioned problems. Such a mixture not only improves the quality of the basa fillet during storage but also provides health benefits for the consumer and a huge opportunity for export earnings with great potential application in the seafood industry to produce further processed fish meat products from high-quality fish fillet.

## Figures and Tables

**Figure 1 foods-12-04170-f001:**
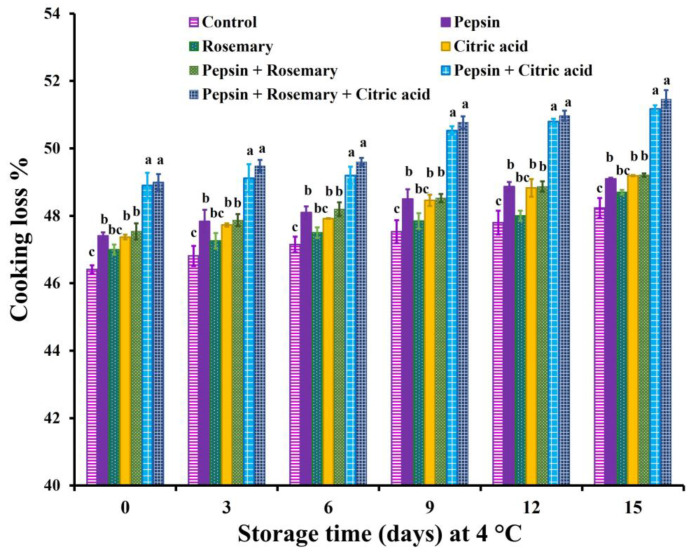
Cooking loss % of basa fish fillets-treated with pepsin enzyme (0.1%), rosemary EOs (0.5%), citric acid (0.5%), and their combination during chilling storage at 4 °C for 15 days. Columns with different letters within the same day of storage are significantly different at *p* < 0.05.

**Figure 2 foods-12-04170-f002:**
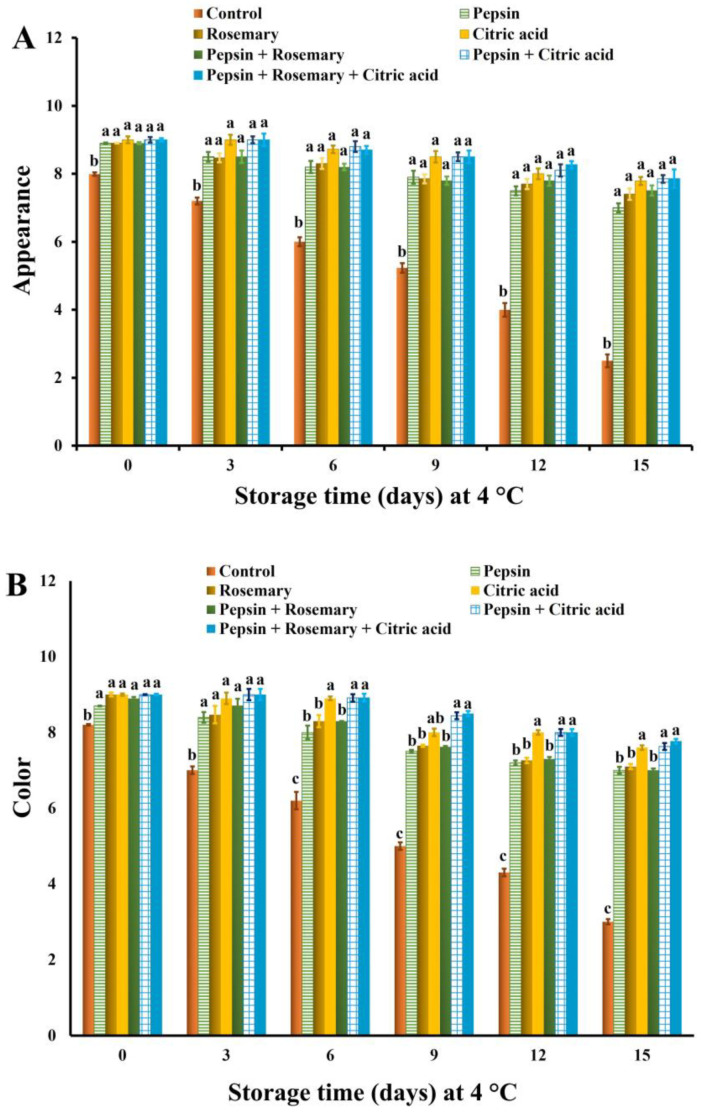
Appearance (**A**) and color (**B**) scores of basa fish fillets treated with pepsin enzyme (0.1%), rosemary EOs (0.5%), citric acid (0.5%), and their combination during chilling storage at 4 °C for 15 days. Columns with different letters within the same day of storage are significantly different at *p* < 0.05.

**Figure 3 foods-12-04170-f003:**
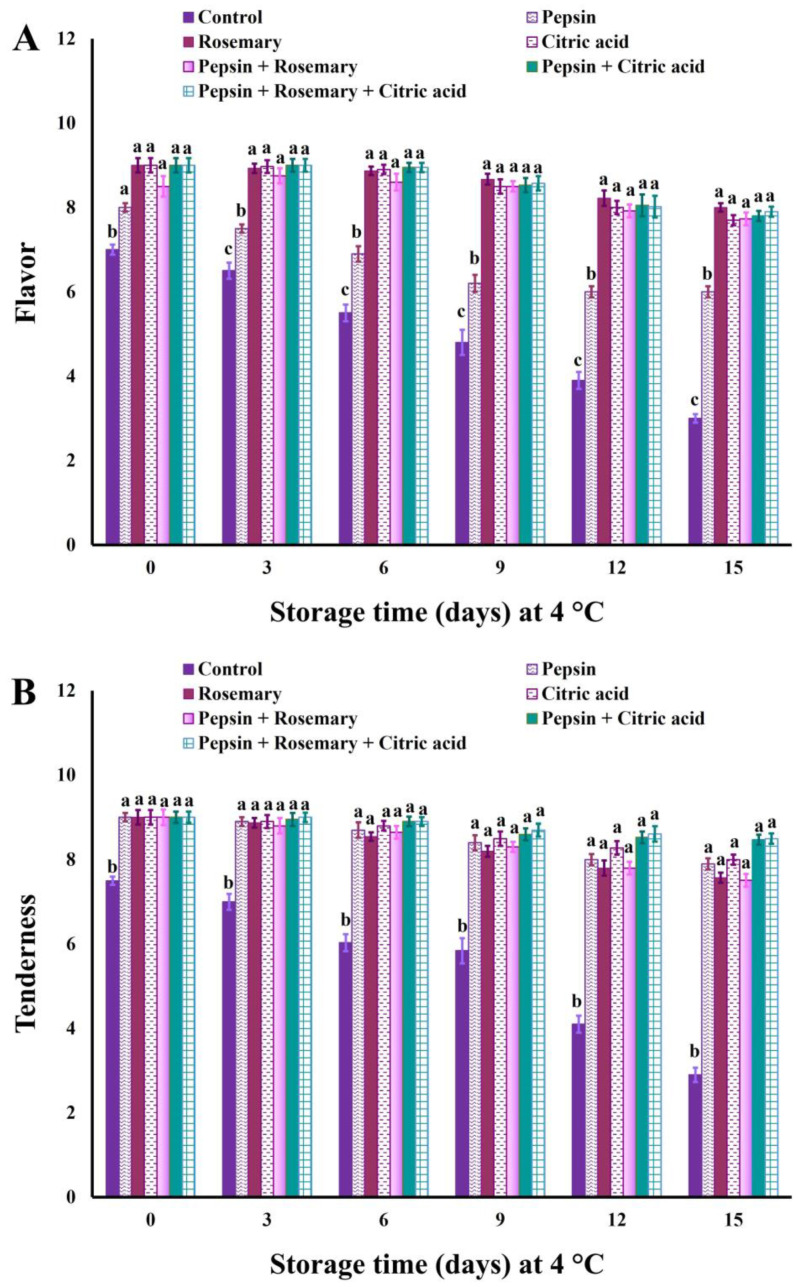
Flavor (**A**) and tenderness (**B**) scores of basa fish fillets treated with pepsin enzyme (0.1%), rosemary EOs (0.5%), citric acid (0.5%), and their combination during chilling storage at 4 °C for 15 days. Columns with different letters within the same day of storage are significantly different at *p* < 0.05.

**Figure 4 foods-12-04170-f004:**
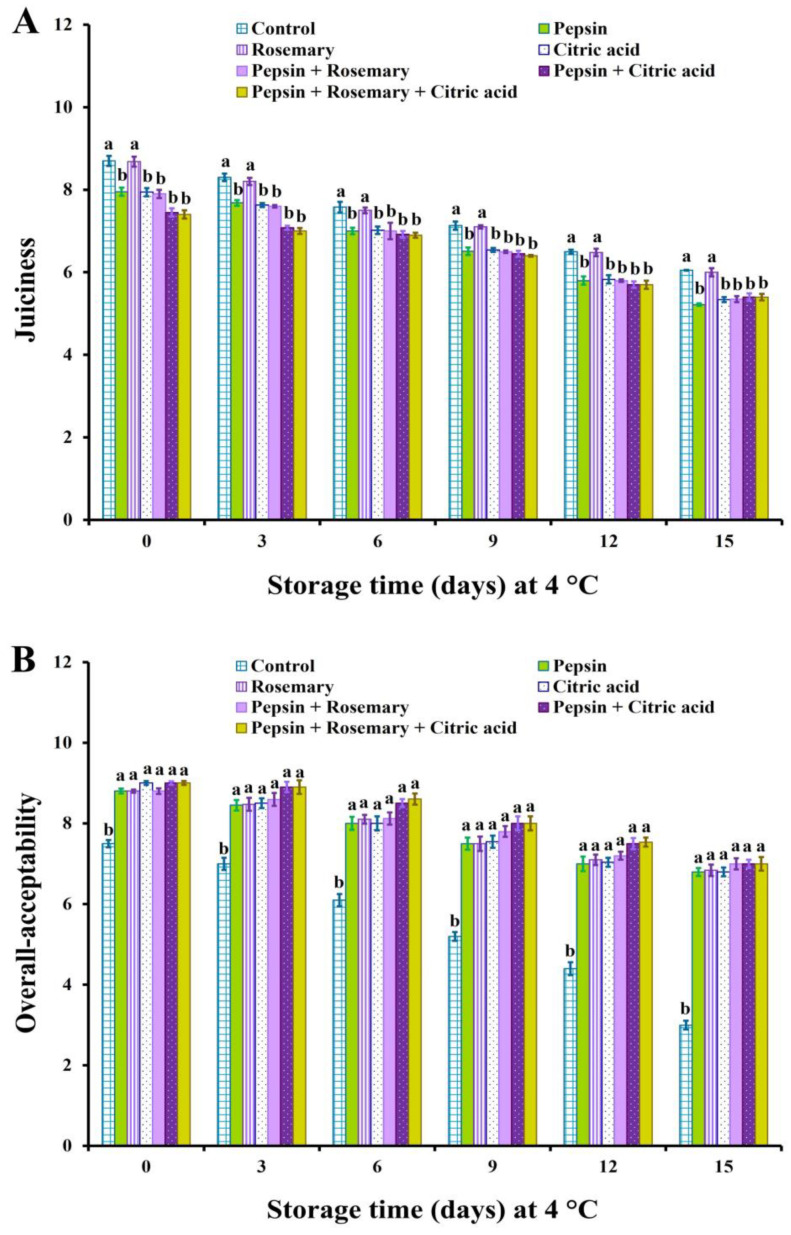
Juiciness (**A**) and overall-acceptability (**B**) scores of basa fish fillets treated with pepsin enzyme (0.1%), rosemary EOs (0.5%), citric acid (0.5%), and their combination during chilling storage at 4 °C for 15 days. Columns with different letters within the same day of storage are significantly different at *p* < 0.05.

**Table 1 foods-12-04170-t001:** Treatment conditions of basa fish fillets.

	Group 1	Group 2	Group 3	Group 4	Group 5	Group 6	Group 7
	Frozen Basa Fillet Samples Were Thawed in a Refrigerator at 4 °C for 12 h						
Control (C)	DW						
Pepsin (E)		0.1%			0.1%	0.1%	0.1%
Rosemary (R)			0.5%		0.5%		0.5%
Citric acid (CA)				0.5%		0.5%	0.5%
	All groups were packed in sterile polyethylene bags with their solutions and stored in a refrigerator at 4 °C for 15 days.						

**Table 2 foods-12-04170-t002:** Enzyme activity, protein content, and specific activity of the pepsin enzyme used in the marination of the basa fish fillet.

Pepsin Enzyme Activity (U/g)	223.34 ± 2.03
Protein (mg/g)	0.0361 ± 0.00
The specific activity of the pepsin enzyme (U/mg)	6186.7 ± 43.40

Values represent the mean of three independent replicates ± SE.

**Table 3 foods-12-04170-t003:** Chemical composition of rosemary (*Rosmarinus officinalis*) EOs used in the marination of the basa fish fillet.

Peak	Component Name	Retention Time (min)	Area Sum %	Area
1	α-Pinene	6.309	10.42	31,497,707.92
2	Camphene	6.675	4.89	14,777,274.78
3	β-Pinene	7.373	0.87	2,639,939.72
4	β-Myrcene	7.694	0.86	2,593,934.97
5	α-Terpinene	8.397	0.44	1,337,322.63
6	p-Cymene	8.621	3.28	9,922,777.81
7	D-Limonene	8.741	2.59	7,840,131.92
8	Eucalyptol	8.849	33.15	100,168,360.3
9	Linalol	10.669	1.14	3,456,902.64
10	Camphor	12.019	18.84	56,937,542.27
11	Isoborneol	12.603	13.53	40,897,717.81
12	Terpinen-4-ol	12.895	1.41	4,253,812.82
13	α-Terpineol	13.273	5.61	16,953,031.88
14	Bornyl acetate	15.888	1.67	5,043,518.7
15	Carvacrol	16.259	0.48	1,445,287.6
16	β-Caryophyllene	19.447	0.80	2,403,027.08

**Table 4 foods-12-04170-t004:** Proximate chemical analysis of basa fish fillets-treated with pepsin enzyme (E, 0.1%), rosemary EO_S_ (R, 0.5%), citric acid (CA, 0.5%), and their combination.

	Moisture (g/100 g)	Protein (g/100 g)	Fat (g/100 g)	Ash (g/100 g)
C	85.03 ^c^ ± 0.07	9.50 ^c^ ± 0.14	3.35 ^a^ ± 0.15	2.11 ^a^ ± 0.07
E	86.58 ^ab^ ± 0.04	11.02 ^ab^ ± 0.14	1.39 ^cd^ ± 0.14	0.99 ^b^ ± 0.08
R	85.19 ^c^ ± 0.39	10.40 ^b^ ± 0.43	2.43 ^b^ ± 0.20	1.95 ^a^ ± 0.08
CA	86.75 ^a^ ± 0.07	10.97 ^a^ ± 0.03	1.29 ^de^ ± 0.07	0.98 ^b^ ± 0.10
E + R	86.08 ^b^ ± 0.12	11.22 ^a^ ± 0.12	1.72 ^c^ ± 0.04	0.96 ^b^ ± 0.04
E + CA	86.95 ^a^ ± 0.19	11.25 ^a^ ± 0.23	1.00 ^e^ ± 0.06	0.79 ^bc^ ± 0.01
E + R + CA	86.86 ^a^ ± 0.16	11.39 ^a^ ± 0.17	1.03 ^de^ ± 0.03	0.70 ^c^ ± 0.10

^a–e^ Means with different superscripts within the same column for each parameter are significantly (*p* < 0.05) different. Values represent the mean of three independent replicates ± SE.

**Table 5 foods-12-04170-t005:** Protein deterioration and fat oxidation parameters of basa fish fillets-treated with pepsin enzyme (E, 0.1%), rosemary EOs (R, 0.5%), citric acid (CA, 0.5%), and their combination during chilling storage at 4 °C for 15 days.

Deterioration Criteria	Treatments	Storage Period (Days)					
		0-Time	3-Day	6-Day	9-Day	12-Day	15-Day
pH	C	6.40 ^a^ ± 0.03	6.56 ^a^ ± 0.03	6.80 ^a^ ± 0.03	6.84 ^a^ ± 0.02	6.94 ^a^ ± 0.01	7.00 ^a^ ± 0.01
	E	5.93 ^c^ ± 0.02	6.11 ^b^ ± 0.05	6.16 ^cd^ ± 0.03	6.25 ^d^ ± 0.02	6.35 ^d^ ± 0.02	6.38 ^de^ ± 0.02
	R	6.01 ^b^ ± 0.01	6.17 ^b^ ± 0.02	6.35 ^b^ ± 0.02	6.51 ^b^ ± 0.01	6.64 ^b^ ± 0.01	6.70 ^b^ ± 0.01
	CA	5.83 ^d^ ± 0.01	5.95 ^c^ ± 0.03	6.12 ^de^ ± 0.04	6.22 ^de^ ± 0.01	6.34 ^d^ ± 0.03	6.42 ^d^ ± 0.01
	E + R	6.00 ^b^ ± 0.00	6.14 ^b^ ± 0.02	6.24 ^c^ ± 0.02	6.35 ^c^ ± 0.02	6.45 ^c^ ± 0.01	6.54 ^c^ ± 0.02
	E + CA	5.75 ^e^ ± 0.01	5.86 ^d^ ± 0.02	6.04 ^e^ ± 0.04	6.14 ^f^ ± 0.01	6.23 ^e^ ± 0.01	6.29 ^f^ ± 0.01
	E + R + CA	5.78 ^de^ ± 0.01	5.92 ^cd^ ± 0.01	6.09 ^de^ ± 0.03	6.19 ^e^ ± 0.01	6.27 ^e^ ± 0.01	6.35 ^e^ ± 0.01
TVBN	C	2.78 ^a^ ± 0.06	3.44 ^a^ ± 0.15	4.17 ^a^ ±0.10	5.02 ^a^ ± 0.13	5.54 ^a^ ± 0.09	6.52 ^a^ ± 0.05
	E	1.26 ^c^ ± 0.02	1.53 ^c^ ± 0.03	1.58 ^c^ ± 0.02	1.94 ^c^ ± 0.02	2.74 ^c^ ± 0.03	3.36 ^c^ ± 0.02
	R	1.38 ^b^ ± 0.02	1.72 ^b^ ± 0.03	2.08 ^b^ ± 0.04	2.58 ^b^ ± 0.03	3.63 ^b^ ± 0.03	4.92 ^b^ ± 0.02
	CA	0.89 ^e^ ± 0.03	1.05 ^e^ ± 0.03	1.23 ^d^ ± 0.02	1.74 ^d^ ± 0.03	2.56 ^d^ ± 0.03	3.20 ^d^ ± 0.02
	E + R	1.07 ^d^ ± 0.03	1.33 ^d^ ± 0.03	1.47 ^c^ ± 0.02	1.76 ^d^ ± 0.02	2.41 ^e^ ± 0.02	3.32 ^c^ ± 0.02
	E + CA	0.78 ^f^ ± 0.03	0.96 ^ef^ ± 0.03	1.17 ^d^ ± 0.02	1.30 ^e^ ± 0.01	1.72 ^f^ ± 0.02	2.14 ^e^ ± 0.01
	E + R + CA	0.66 ^g^ ± 0.02	0.82 ^f^ ± 0.01	1.10 ^d^ ± 0.01	1.26 ^e^ ± 0.01	1.63 ^f^ ± 0.01	2.04 ^f^ ± 0.02
TBA	C	0.58 ^a^ ± 0.01	0.83 ^a^ ± 0.01	1.09 ^a^ ± 0.01	1.61 ^a^ ± 0.01	2.00 ^a^ ± 0.01	2.68 ^a^ ± 0.01
	E	0.32 ^b^ ± 0.01	0.36 ^c^ ± 0.01	0.39 ^c^ ± 0.01	0.49 ^c^ ± 0.01	0.56 ^c^ ±0.01	0.70 ^c^ ± 0.01
	R	0.33 ^b^ ± 0.01	0.56 ^b^ ± 0.01	0.57 ^b^ ± 0.01	0.66 ^b^ ± 0.01	0.74 ^b^ ± 0.01	0.88 ^b^ ± 0.01
	CA	0.16 ^d^ ± 0.01	0.18 ^e^ ± 0.01	0.36 ^c^ ± 0.01	0.41 ^d^ ± 0.01	0.44 ^e^ ± 0.01	0.57 ^e^ ± 0.01
	E + R	0.25 ^c^ ± 0.01	0.32 ^d^ ± 0.01	0.36 ^c^ ± 0.01	0.46 ^c^ ± 0.01	0.49 ^d^ ± 0.01	0.65 ^d^ ± 0.03
	E + CA	0.14 ^d^ ± 0.01	0.18 ^e^ ± 0.01	0.26 ^d^ ± 0.01	0.32 ^e^ ± 0.01	0.39 ^f^ ± 0.01	0.44 ^f^ ± 0.01
	E + R + CA	0.11 ^e^ ± 0.01	0.16 ^e^ ± 0.01	0.20 ^d^ ± 0.01	0.29 ^e^ ±0.01	0.36 ^f^ ± 0.01	0.40 ^f^ ± 0.01
FFAs	C	0.73 ^a^ ± 0.02	0.97 ^a^ ± 0.01	1.43 ^a^ ± 0.10	1.75 ^a^ ± 0.03	1.94 ^a^ ± 0.02	2.46 ^a^ ± 0.03
	E	0.47 ^c^ ± 0.04	0.54 ^c^ ± 0.01	0.62 ^b^ ± 0.02	0.74 ^b^ ± 0.02	0.83 ^bc^ ±0.03	0.94 ^b^ ± 0.01
	R	0.53 ^b^ ± 0.02	0.57 ^b^ ± 0.01	0.70 ^b^ ± 0.01	0.75 ^b^ ± 0.02	0.87 ^b^ ± 0.01	0.95 ^b^ ± 0.01
	CA	0.21 ^d^ ± 0.01	0.27 ^d^ ± 0.01	0.31 ^c^ ± 0.01	0.33 ^c^ ± 0.02	0.42 ^d^ ± 0.02	0.52 ^c^ ± 0.01
	E + R	0.44 ^c^ ± 0.03	0.51 ^c^ ± 0.01	0.58 ^b^ ± 0.02	0.71 ^b^ ± 0.01	0.81 ^c^ ± 0.02	0.91 ^b^ ± 0.01
	E + CA	0.16 ^de^ ± 0.01	0.17 ^e^ ± 0.00	0.25 ^c^ ± 0.01	0.30 ^cd^ ± 0.00	0.38 ^d^ ± 0.01	0.44 ^d^ ± 0.02
	E + R + CA	0.12 ^e^ ± 0.01	0.13 ^f^ ± 0.00	0.20 ^c^ ± 0.01	0.26 ^d^ ± 0.01	0.37 ^d^ ± 0.01	0.42 ^d^ ± 0.01
AN	C	1.15 ^a^ ± 0.03	1.39 ^a^ ± 0.01	1.54 ^a^ ± 0.02	1.75 ^a^ ± 0.03	2.17 ^a^ ± 0.02	2.92 ^a^ ± 0.02
	E	0.88 ^c^ ± 0.04	0.93 ^c^ ± 0.02	0.98 ^c^ ± 0.01	1.12 ^c^ ± 0.02	1.26 ^c^ ± 0.02	1.35 ^b^ ± 0.01
	R	0.95 ^b^ ± 0.01	1.01 ^b^ ± 0.02	1.17 ^b^ ± 0.02	1.26 ^b^ ± 0.02	1.38 ^b^ ± 0.01	1.46 ^c^ ± 0.01
	CA	0.65 ^e^ ± 0.02	0.67 ^e^ ± 0.01	0.74 ^e^ ± 0.02	0.80 ^e^ ± 0.02	0.84 ^e^ ± 0.01	0.97 ^e^ ± 0.01
	E + R	0.75 ^d^ ± 0.03	0.83 ^d^ ± 0.02	0.90 ^d^ ± 0.03	0.95 ^d^ ± 0.03	1.17 ^d^ ± 0.02	1.22 ^d^ ± 0.01
	E + CA	0.57 ^f^ ± 0.01	0.59 ^f^ ± 0.01	0.65 ^f^ ± 0.03	0.76 ^e^ ± 0.03	0.83 ^e^ ±0.01	0.95 ^e^ ± 0.01
	E + R + CA	0.52 ^f^ ± 0.01	0.54 ^f^ ± 0.01	0.63 ^f^ ± 0.03	0.75 ^e^ ± 0.03	0.81 ^e^ ± 0.01	0.94 ^e^ ± 0.02

^a–g^ Means with different superscripts within the same column for each parameter are significantly (*p* < 0.05) different. Values represent the mean of three independent replicates ± SE. TVBN: total volatile base nitrogen (mg %); TBA: thiobarbituric acid (mg malonaldehyde/Kg); FFA: free fatty acid % as Oleic acid; AN: acid number (mg NaOH/g).

**Table 6 foods-12-04170-t006:** Color and shear force values of basa fish fillets-treated with pepsin enzyme (0.1%), rosemary EO_S_ (0.5%), citric acid (0.5%), and their combination.

Treatments	Color				Shear Force (Kgf)
	L*	a*	b*	WI	
C	61.01 ^de^ ± 0.70	1.26 ^d^ ± 0.12	5.22 ^a^ ± 0.06	60.63 ^de^ ± 0.66	19.51 ^a^ ± 0.26
E	58.80 ^e^ ± 0.29	1.48 ^cd^ ± 0.10	3.17 ^b^ ± 0.13	58.65 ^e^ ± 0.25	12.00 ^d^ ± 0.12
R	62.83 ^cd^ ± 0.61	1.28 ^d^ ± 0.05	3.02 ^b^ ± 0.08	62.67 ^cd^ ± 0.55	15.09 ^b^ ± 0.07
CA	65.46 ^ab^ ± 0.65	2.12 ^bc^ ± 0.13	2.08 ^c^ ± 0.01	65.34 ^ab^ ± 0.63	13.38 ^c^ ± 0.31
E + R	62.80 ^cd^ ± 0.71	1.77 ^bd^ ± 0.32	3.00 ^b^ ± 0.04	62.62 ^cd^ ± 0.67	10.32 ^e^ ± 0.13
E + CA	64.07 ^bc^ ± 0.45	2.37 ^ab^ ± 0.17	2.05 ^c^ ± 0.05	63.92 ^bc^ ± 0.40	9.64 ^f^ ± 0.05
E + R + CA	67.42 ^a^ ± 1.33	3.07 ^a^ ± 0.54	2.00 ^c^ ± 0.07	67.20 ^a^ ± 1.20	8.58 ^g^ ± 0.15

^a–g^ Means with different superscripts within the same column for each parameter are significantly (*p* < 0.05) different. Values represent the mean of three independent replicates ± SE. L*: lightness; a*: redness; b*: yellowness; WI: whitness index.

**Table 7 foods-12-04170-t007:** Fatty acid profile (g/100 g fatty acids) of basa fish fillets-treated with pepsin enzyme (E, 0.1%), rosemary EOs (R, 0.5%), citric acid (CA, 0.5%), and their combination during chilling storage at 4 °C for 15 days.

	C	E	R	CA	E + R	E + CA	E + R + CA
C8:0	0.53 ^a^ ± 0.01	0.00 ^b^ ± 0.00	0.00 ^b^ ± 0.00	0.00 ^b^ ± 0.00	0.00 ^b^ ± 0.00	0.00 ^b^ ± 0.00	0.00 ^b^ ± 0.00
C10:0	0.53 ^a^ ± 0.02	0.00 ^c^ ± 0.00	0.44 ^b^ ± 0.02	0.00 ^c^ ± 0.00	0.00 ^c^ ± 0.00	0.00 ^c^ ± 0.00	0.00 ^c^ ± 0.00
C12:0	0.81 ^a^ ± 0.01	0.58 ^c^ ± 0.01	0.66 ^b^ ± 0.02	0.26 ^e^ ± 0.01	0.37 ^d^ ± 0.01	0.00 ^f^ ± 0.00	0.00 ^f^ ± 0.00
C14:0	12.97 ^a^ ± 0.02	4.37 ^c^ ± 0.01	4.59 ^b^ ± 0.02	3.62 ^e^ ± 0.02	3.83 ^d^ ± 0.02	3.47 ^f^ ± 0.01	3.14 ^g^ ± 0.02
C16:0	31.47 ^a^ ± 0.02	30.64 ^c^ ± 0.02	31.05 ^b^ ± 0.03	29.55 ^e^ ± 0.03	30.12 ^d^ ± 0.02	28.01 ^f^ ± 0.01	25.22 ^g^ ± 0.01
C18:0	11.82 ^a^ ± 0.02	11.01 ^c^ ± 0.01	11.27 ^b^ ± 0.01	10.03 ^e^ ± 0.01	10.88 ^d^ ± 0.01	9.04 ^f^ ± 0.02	8.90 ^g^ ± 0.08
C20:0	4.98 ^a^ ± 0.02	1.02 ^c^ ± 0.02	1.61 ^b^ ± 0.01	0.52 ^e^ ± 0.01	0.83 ^d^ ± 0.01	0.24 ^f^ ± 0.01	0.12 ^g^ ± 0.02
C21:0	0.96 ^a^ ± 0.01	0.36 ^c^ ± 0.01	0.44 ^b^ ± 0.02	0.00 ^d^ ± 0.00	0.00 ^d^ ± 0.00	0.00 ^d^ ± 0.00	0.00 ^d^ ± 0.00
SFAs	64.07 ^a^ ± 0.04	47.98 ^c^ ± 0.02	50.06 ^b^ ± 0.03	43.98 ^e^ ± 0.02	46.03 ^d^ ± 0.05	40.76 ^f^ ± 0.03	37.38 ^g^ ± 0.01
C14:1	0.00 ^c^ ± 0.00	0.00 ^c^ ± 0.00	0.00 ^c^ ± 0.00	0.41 ^b^ ± 0.01	0.00 ^c^ ± 0.00	0.44 ^b^ ± 0.01	0.58 ^a^ ± 0.01
C16:1	0.97 ^e^ ± 0.01	1.24 ^d^ ± 0.02	1.23 ^d^ ± 0.02	1.43 ^c^ ± 0.02	1.42 ^c^ ± 0.02	2.14 ^b^ ± 0.02	3.50 ^a^ ± 0.02
C16:1, n7	0.00 ^b^ ± 0.00	0.00 ^b^ ± 0.00	0.00 ^b^ ± 0.00	0.00 ^b^ ± 0.00	0.00 ^b^ ± 0.00	0.00 ^b^ ± 0.00	0.42 ^a^ ± 0.02
C18:1n9c	26.18 ^g^ ± 0.02	36.13 ^e^ ± 0.01	35.58 ^f^ ± 0.01	37.03 ^c^ ± 0.03	36.96 ^d^ ± 0.02	37.34 ^b^ ± 0.02	37.53 ^a^ ± 0.02
C20:1	0.00 ^f^ ± 0.00	1.97 ^d^ ± 0.01	1.78 ^e^ ± 0.01	2.13 ^c^ ± 0.01	2.12 ^c^ ± 0.03	2.98 ^b^ ± 0.01	3.74 ^a^ ± 0.02
C20:1, trans	0.55 ^a^ ± 0.01	0.41 ^c^ ± 0.02	0.47 ^b^ ± 0.01	0.00 ^d^ ± 0.00	0.00 ^d^ ± 0.00	0.00 ^d^ ± 0.00	0.00 ^d^ ± 0.00
MUFAs	27.70 ^f^ ± 0.04	39.75 ^d^ ±0.02	39.06 ^e^ ± 0.01	41.00 ^c^ ± 0.01	40.50 ^c^ ± 0.04	42.90 ^b^ ± 0.01	45.77 ^a^ ±0.06
C18:2n6c	7.08 ^f^ ± 0.02	9.54 ^d^ ± 0.01	9.03 ^e^ ± 0.02	12.02 ^b^ ± 0.01	10.75 ^c^ ± 0.03	12.75 ^a^ ± 0.02	12.78 ^a^ ± 0.01
C18:2n6t	0.45 ^a^ ± 0.01	0.00 ^b^ ± 0.00	0.00 ^b^ ± 0.00	0.00 ^b^ ± 0.00	0.00 ^b^ ± 0.00	0.00 ^b^ ± 0.00	0.00 ^b^ ± 0.00
C18:3n6 CLA	0.00 ^d^ ± 0.00	0.60 ^b^ ± 0.00	0.13 ^c^ ± 0.01	0.70 ^a^ ± 0.03	0.61 ^b^ ± 0.02	0.71 ^a^ ± 0.03	0.76 ^a^ ± 0.02
C18:3n3	0.00 ^e^ ± 0.00	1.33 ^c^ ± 0.02	0.92 ^d^ ± 0.01	1.33 ^c^ ± 0.01	1.29 ^c^ ± 0.01	1.55 ^b^ ± 0.02	1.98 ^a^ ± 0.01
C20:5n-3	0.70 ^d^ ± 0.00	0.80 ^c^ ± 0.01	0.80 ^c^ ± 0.00	0.97 ^b^ ± 0.02	0.82 ^c^ ± 0.01	1.33 ^a^ ± 0.02	1.33 ^a^ ± 0.01
PUFAs	8.23 ^g^ ± 0.01	12.27 ^e^ ± 0.03	10.88 ^f^ ± 0.06	15.02 ^c^ ± 0.02	13.47 ^d^ ± 0.01	16.34 ^b^ ± 0.01	16.85 ^a^ ± 0.02
UFAs	35.93 ^g^ ± 0.04	52.02 ^e^ ± 0.03	49.94 ^f^ ± 0.01	56.02 ^c^ ± 0.01	53.97 ^d^ ± 0.02	59.24 ^b^ ± 0.03	62.62 ^a^ ± 0.06
PUFAs/SFAs	0.13 ^e^ ± 0.03	0.26 ^c^ ± 0.01	0.22 ^d^ ± 0.00	0.34 ^b^ ± 0.00	0.29 ^bc^ ± 0.03	0.41 ^a^ ± 0.04	0.45 ^a^ ± 0.03
n-6/n-3	10.76 ^a^ ± 0.06	4.76 ^e^ ± 0.02	5.32 ^d^ ± 0.01	5.53 ^b^ ± 0.02	5.38 ^c^ ± 0.01	4.67 ^f^ ± 0.01	4.09 ^g^ ± 0.03
n-3/n-6	0.10 ^c^ ± 0.02	0.21 ^ab^ ± 0.03	0.18 ^b^ ± 0.05	0.19 ^ab^ ± 0.01	0.19 ^ab^ ± 0.01	0.21 ^ab^ ± 0.02	0.24 ^a^ ± 0.03
AI	2.34 ^a^ ± 0.04	0.94 ^c^ ± 0.01	1.00 ^b^ ± 0.02	0.79 ^d^ ± 0.05	0.84 ^d^ ± 0.07	0.71 ^e^ ± 0.02	0.60 ^f^ ± 0.01
TI	2.84 ^a^ ± 0.03	1.46 ^c^ ± 0.01	1.59 ^b^ ± 0.04	1.27 ^e^ ± 0.01	1.38 ^d^ ± 0.02	1.09 ^f^ ± 0.02	0.94 ^g^ ± 0.01
NVI	1.21 ^e^ ± 0.01	1.54 ^d^ ± 0.03	1.51 ^d^ ± 0.02	1.59 ^c^ ± 0.02	1.59 ^c^ ± 0.01	1.66 ^b^ ± 0.03	1.84 ^a^ ± 0.02

^a–g^ Means with different superscripts within the same row for each parameter are significantly (*p* < 0.05) different. Values represent the mean of three independent replicates ± SE. SFAs: total saturated fatty acids; MUFAs: monounsaturated fatty acids; CLA: conjugated linoleic acid; PUFAs: polyunsaturated fatty acids; UFAs: total unsaturated fatty acids; AI: atherogenic index; TI: thrombogenic index; NVI: nutritive value index.

**Table 8 foods-12-04170-t008:** Microbial counts (Log_10_ CFU/g) of basa fish fillets-treated with pepsin enzyme (E, 0.1%), rosemary EOs (R, 0.5%), citric acid (CA, 0.5%), and their combination during chilling storage at 4 °C for 15 days.

Microbial Category	Treatments	Storage Period (Days)					
		0-Time	3-Day	6-Day	9-Day	12-Day	15-Day
APC	C	3.73 ^a^ ± 0.08	4.05 ^a^ ± 0.13	5.10 ^a^ ± 0.08	5.14 ^a^ ± 0.20	7.01 ^a^ ± 0.17	7.58 ^a^ ± 0.07
	E	<2 ^b^ ± 0.00	<2 ^b^ ± 0.00	2.25 ^c^ ±0.07	3.03 ^bc^ ± 0.15	3.83 ^c^ ± 0.17	4.46 ^bc^ ±0.15
	R	<2 ^b^ ± 0.00	<2 ^b^ ± 0.00	2.80 ^b^ ± 0.06	3.33 ^b^ ± 0.21	4.28 ^b^ ± 0.11	4.71 ^b^ ±0.06
	CA	<2 ^b^ ± 0.00	<2 ^b^ ± 0.00	2.17 ^cd^ ± 0.09	2.83 ^c^ ± 0.09	3.70 ^cd^ ± 0.10	4.33 ^cd^ ±0.09
	E + R	<2 ^b^ ± 0.00	<2 ^b^ ± 0.00	2.04 ^d^ ± 0.04	2.67 ^c^ ±0.12	3.38 ^d^ ± 0.19	4.04 ^d^ ±0.04
	E + CA	<2 ^b^ ± 0.00	<2 ^b^ ± 0.00	<2 ^e^ ± 0.00	<2 ^d^ ± 0.00	2.53 ^e^ ± 0.04	3.50 ^e^ ± 0.14
	E + R + CA	<2 ^b^ ± 0.00	<2 ^b^ ± 0.00	<2 ^e^ ± 0.00	<2 ^d^ ± 0.00	2.14 ^e^ ± 0.09	3.26 ^e^ ± 0.14
Psychrotrophs	C	2.83 ^a^ ± 0.17	3.35 ^a^ ± 0.05	3.78 ^a^ ± 0.06	4.73 ^a^ ± 0.12	5.03 ^a^ ± 0.03	6.33 ^a^ ± 0.17
	E	<2 ^b^ ± 0.00	<2 ^b^ ± 0.00	<2 ^b^ ± 0.00	2.13 ^c^ ± 0.09	3.10 ^b^ ± 0.15	3.92 ^bc^ ±0.04
	R	<2 ^b^ ± 0.00	<2 ^b^ ± 0.00	<2 ^b^ ± 0.00	2.44 ^b^ ±0.10	3.27 ^b^ ± 0.23	4.11 ^b^ ± 0.21
	CA	<2 ^b^ ± 0.00	<2 ^b^ ± 0.00	<2 ^b^ ± 0.00	2.00 ^c^ ±0.00	2.68 ^c^ ± 0.08	3.82 ^bc^ ± 0.04
	E + R	<2 ^b^ ± 0.00	<2 ^b^ ± 0.00	<2 ^b^ ± 0.00	<2 ^d^ ± 0.00	2.58 ^cd^ ± 0.04	3.62 ^cd^ ± 0.12
	E + CA	<2 ^b^ ± 0.00	<2 ^b^ ± 0.00	<2 ^b^ ± 0.00	<2 ^d^ ± 0.00	2.32 ^de^ ±0.07	3.55 ^cd^ ± 0.10
	E + R + CA	<2 ^b^ ± 0.00	<2 ^b^ ± 0.00	<2 ^b^ ± 0.00	<2 ^d^ ± 0.00	2.18 ^e^ ±0.10	3.25 ^d^ ± 0.09
Enterobacteriaceae	C	2.57 ^a^ ± 0.18	3.16 ^a^ ± 0.09	4.36 ^a^ ± 0.23	5.53 ^a^ ± 0.27	6.07 ^a^ ± 0.12	7.06 ^a^ ± 0.03
	E	<2 ^b^ ± 0.00	<2 ^b^ ± 0.00	<2 ^b^ ± 0.00	2.63 ^bc^ ± 0.12	3.77 ^b^ ± 0.11	4.43 ^c^ ± 0.22
	R	<2 ^b^ ± 0.00	<2 ^b^ ± 0.00	<2 ^b^ ± 0.00	3.03 ^b^ ± 0.15	4.00 ^b^ ± 0.09	4.86 ^b^ ± 0.03
	CA	<2 ^b^ ± 0.00	<2 ^b^ ± 0.00	<2 ^b^ ± 0.00	2.35 ^c^ ± 0.13	3.69 ^bc^ ± 0.12	4.39 ^cd^ ± 0.16
	E + R	<2 ^b^ ± 0.00	<2 ^b^ ± 0.00	<2 ^b^ ± 0.00	<2 ^d^ ± 0.00	3.35 ^cd^ ± 0.05	4.20 ^ce^ ± 0.06
	E + CA	<2 ^b^ ± 0.00	<2 ^b^ ± 0.00	<2 ^b^ ± 0.00	<2 ^d^ ± 0.00	3.09 ^d^ ± 0.09	4.07 ^de^ ± 0.12
	E + R + CA	<2 ^b^ ± 0.00	<2 ^b^ ± 0.00	<2 ^b^ ± 0.00	<2 ^d^ ± 0.00	2.50 ^e^ ±0.20	3.87 ^e^ ± 0.09

^a–e^ Means with different superscripts within the same column for each parameter are significantly (*p* < 0.05 or *p* < 0.01) different. Values represent the mean of three independent replicates ± SE.

## Data Availability

Data are contained within the article.

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
