# Peer review of "Improvement of Microbial Quality, Physicochemical Properties, Fatty Acids Profile, and Shelf Life of Basa (Pangasius bocourti) Fillets during Chilling Storage Using Pepsin, Rosemary Oil, and Citric Acid"

_foods, 2023, doi:10.3390/foods12224170_

Round 1
Reviewer 1 Report
Comments and Suggestions for Authors
This study aim to use pepsin, rosemary oil and citric acid to improve quality and shelf-life of fish fillets. After carefully examining your manuscript, I think this MS needs to be largely re-worked. The structure of experiments and results was confusing. It is too long. Sentences should be simplified (lots of redundancy). Results should not be repeated again and again. The findings of the study are fairly discussed. The discussion related to theory and other previous research should add to strengthen scientific soundness.
Here is some comment.
- I suggested shorten the topic by combining some parameters and changes into the words that covered them such as (lipid stability, fatty acid profile = lipid profiles) or changed into “Improvement of physicochemical quality, microbial population and shelf-life of……..”, etc. Moreover, using “pepsin” is enough, no need to use “pepsin enzyme”. Also, which part of rose mary? Oil? Please check the topic again!
ABSTRACT
- line26: the general information of pepsin, rosemary oil and citric acid such as concentration, should be added in the abstract. Also, their combination? All three? How’s about only two. This is unclear.
- line27: Author reported that the protein content increased in all sample!!!! Is it significantly increased? Please check, since it seem like impossible or very hard to increase the protein content of fish by adding some additives!!!!!!
INTRODUCTION
- Information related to the selected conditions/concentration used for each factors cited from previous studies should be addressed to support the design of experiment in this study.
- Moreover, information based on in-depth how’s each factors like pepsin can retard the spoilage or improve the changes in quality during storage should be addressed as well.
- Moreover, the reasons why author design to use them in combination should be added. Are they provide the synergistic effect based on previous review, etc. Did previous studies used them together? If had, what’s are the new findings of this study, etc. This information should be appeared in the introduction to increase scientific introduction to the research.
MATERIAL AND METHODS
- Why should measure the pepsin activity? Please describe in the aspect of quality preservation. For the experimental design, adding pepsin accelerate or retard the spoilage of the fish sample? Please clarify this point.
- Based on pepsin is generally accumulated in gut/viscera, which was removed from the fillet? So, why should determine pepsin activity? If author want to monitor the enzymatic ac. Why not determine protease ac. Or other which can refer to all enzymes not only pepsin?
RESULTS AND DISCUSSION
- Why should author need to determine pepsin activity?
- For the rosemary oil compounds, please increase how’s each compounds (only main compounds or crucial is enough) affect to the fish quality.
- The discussion of each parameter should be added. Now, it seem like author only report the results.
- Why protein content of fish increase after treated with additives? Please add the discussion.
- Did proximate composition measure only on day0? Why? How to prepare the samples? Treat with the fish and then measure? If like this, why is highly impact to change the composition?
- Why author did not measure the proximate compositions during storage?
- The results were a little bit surprise, for example, SD. appeared at the low and equal level (Table4, all S.D. are only 0.01-0.03?)
- The units of each parameters did not appear.
- The citation with the standard did not provide likes how much perception of TVB-N in fish fillet? How much microorganisms that confirmed the fish is still safe, etc.
- Why author did not measure the pathogen as per standard required since this experiment aimed to evaluate the shelf-life of the product.
Etc.
Author Response
Reviewer≠1
This study aim to use pepsin, rosemary oil and citric acid to improve quality and shelf-life of fish fillets. After carefully examining your manuscript, I think this MS needs to be largely re-worked. The structure of experiments and results was confusing. It is too long. Sentences should be simplified (lots of redundancy). Results should not be repeated again and again. The findings of the study are fairly discussed. The discussion related to theory and other previous research should add to strengthen scientific soundness.
Dear Reviewer,
Thank you for your efforts in reviewing our manuscript and your valuable comments which have greatly contributed to increasing its quality! Please read below our answers to the raised concerns.
Here is some comment.
- I suggested shorten the topic by combining some parameters and changes into the words that covered them such as (lipid stability, fatty acid profile = lipid profiles) or changed into “Improvement of physicochemical quality, microbial population and shelf-life of……..”, etc. Moreover, using “pepsin” is enough, no need to use “pepsin enzyme”. Also, which part of rose mary? Oil? Please check the topic again!
Answer: Special thanks for this suggestion! The topic was shortened, the word enzyme was deleted, and the word oil was added after rosemary. The new title is “Improvement of microbial quality, physicochemical properties, fatty acids profile, and shelf life of basa (Pangasius bocourti) fillets during chilling storage using pepsin, rosemary oil, and citric acid“. Additionally, the sentence on pepsin enzyme activity was deleted.
ABSTRACT
- line26: the general information of pepsin, rosemary oil and citric acid such as concentration, should be added in the abstract. Also, their combination? All three? How’s about only two. This is unclear.
Answer: thank you so much for your comment. All required information was added (lines 29-30 in the abstract).
- line27: Author reported that the protein content increased in all sample!!!! Is it significantly increased? Please check, since it seem like impossible or very hard to increase the protein content of fish by adding some additives!!!!!!
Answer: The addition of pepsin, rosemary oil, and citric acid, in the present study, significantly increases the protein content of treated fish fillet owing to the proteolytic effect of pepsin (Table 2) and their low pH values (Table 5) with subsequent increasing protein dissolution and soluble protein content. Such clarification was added in the discussion section (lines 366-369). This observation was also reported by Gu et al. (2021), who found that both pH and ion strength play critical roles in protein dissolution and denaturation. Similarly, a significant increase in protein and moisture content with a significant reduction in fat content in spent hen nuggets treated with kiwi (containing actinidin proteolytic enzymes) was reported by Pooona et al. (2019). Moreover, Cardinali et al. (2015) reported that rosemary supplementation improved the protein content of rabbit meat.
Reference:
Gu, Z., Liu, S., Duan, Z., Kang, R., Zhao, M., Xia, G., & Shen, X. (2021). Effect of citric acid on physicochemical properties and protein structure of low‐salt restructured tilapia (Oreochromis mossambicus) meat products. Journal of the Science of Food and Agriculture, 101(4), 1636-1645.
Pooona J, Singh P, Prabhakaran P (2019). Effect of kiwifruit
juice and tumbling on tenderness and lipid oxidation in chicken nuggets. Nutr. Food Sci., 50(1): 74‒83.
Cardinali, R., Cullere, M., Dal Bosco, A., Mugnai, C., Ruggeri, S., Mattioli, S., ... & Dalle Zotte, A. (2015). Oregano, rosemary and vitamin E dietary supplementation in growing rabbits: Effect on growth performance, carcass traits, bone development and meat chemical composition. Livestock Science, 175, 83-89.
INTRODUCTION
- Information related to the selected conditions/concentration used for each factors cited from previous studies should be addressed to support the design of experiment in this study.
Answer: An initial experiment was conducted to choose the best concentration of pepsin, rosemary oil, and citric acid on the quality parameters of basa fish fillet. This sentence was added in the materials and methods section (lines 129-131). Additionally, the most recent study published in Foods Journal revealed that using citric acid with a concentration of 2.5% and 5% negatively affects the quality of frozen fish fillets in terms of cooking yield (Klinmalai et al. 2021), which has been already mentioned in the discussion section (lines 525-526). It is noteworthy that no information is yet available about the effect of using pepsin enzyme on the quality of fish fillets (already mentioned in the introduction section). Therefore, we relied on the best concentration selected from our pilot study.
Reference:
Klinmalai, P.; Fong-in, S.; Phongthai, S.; Klunklin, W. Improving the quality of frozen fillets of semi-dried gourami fish (Trichogaster pectoralis) by using sorbitol and citric acid. Foods 2021,
- Moreover, information based on in-depth how’s each factors like pepsin can retard the spoilage or improve the changes in quality during storage should be addressed as well.
Answer: The following sentences explain how pepsin, rosemary, and citric acid improve the quality parameters of basa fish fillet.
The explanation regarding their positive effect on TVBN value and the microbial population: Our result with respect to the significant decrease in TVBN of citric acid and pepsin-treated samples may be due to their low pH value which retard the growth of spoilage bacteria and accumulation of alkaline products such as ammonia. However, the significant decrease of TVBN in rosemary-treated samples may be attributed to the presence of essential compounds that possess antimicrobial and antioxidant activities (Table 3). This clarification has been already mentioned in lines 426-430.
Improving the microbial quality of basa fish fillets, in the present study, using rosemary EOs, citric acid, or pepsin enzyme may be due to their low pH value. Likewise, Brul and Coote [70] attributed the antimicrobial activity of citric acid to the ability of the uncharged form of this acid to penetrate through the microbial cell membrane and acidify its cytoplasm. Furthermore, rosemary EOs are rich in phenolic compounds which possess antibacterial activity. This clarification has been already mentioned in lines 622-627.
The explanation regarding their positive effect on TBA: The obtained low TBA value in the citric acid-treated sample may be due to its ability to chelate prooxidant metals, forming a thermodynamically stable complex consequently lowering their redox potentials. However, improving the lipid stability in rosemary-treated samples is due to their phenolic content which possesses antioxidant activity. In addition, the significant decrease in the TBA value of pepsin-treated samples is linked to its antimicrobial activity arising from its low pH, which could reduce or eliminate microbial rancidity. This clarification has been already mentioned in lines 449-455
The explanation regarding their positive effect on shear force value: Meat tenderness is closely linked to the shear force value, which influences the meat quality and consumer preference for this meat [58,59]. In the present study, shear force values were significantly (P < 0.05) decreased in all treated samples as compared to control ones (Table 5). Such observation may be attributed to their low pH values which causes protein denaturation and weakening of connective tissue. This clarification has been already mentioned in lines 498-501.
- Moreover, the reasons why author design to use them in combination should be added. Are they provide the synergistic effect based on previous review, etc. Did previous studies used them together? If had, what’s are the new findings of this study, etc. This information should be appeared in the introduction to increase scientific introduction to the research.
Answer: Thank you so much for your comment. There are no previous studies that use this combination, which is the novel point in our study. Moreover, a new sentence of novelty point was added in the introduction section (lines 112-114). We use this combination to solve the problems associated with the storage of basa fish fillets (mentioned in the abstract section lines 23-25, introduction section lines 68-72, and conclusion section lines 695-697). The major quality issues associated with the storage of basa fish fillets are meat discoloration, lipid oxidation, and undesirable texture which in turn limits their exportation as well as decreases consumer acceptability. Based on the obtained promising results, using this mixture is considered a good choice to overcome all aforementioned problems. Such a mixture not only improves the quality of basa fillet during storage but also provides health benefits for the consumer and a huge opportunity for export earnings with great potential application in the seafood industry to produce further processed fish meat products from high-quality fish fillet.
MATERIAL AND METHODS
- Why should measure the pepsin activity? Please describe in the aspect of quality preservation. For the experimental design, adding pepsin accelerate or retard the spoilage of the fish sample? Please clarify this point. - Based on pepsin is generally accumulated in gut/viscera, which was removed from the fillet? So, why should determine pepsin activity? If author want to monitor the enzymatic ac. Why not determine protease ac. Or other which can refer to all enzymes not only pepsin?
Answer: Pepsin activity was measured to investigate the potency of this enzyme toward meat protein with subsequent its effect on the texture of fish fillet samples. Adding pepsin retard the spoilage of the fish sample due to low pH value as well as decrease TVBN. Therefore, pepsin could improve microbial quality and sensory quality of basa fish fillets accompanied by an extension in their shelf life during chilling storage at 4 °C for more than 15 days. The application of pepsin herein is a direct addition into basa fish fillet samples and not by feeding for fish after that examine its effect on the quality of fillets.
RESULTS AND DISCUSSION
- Why should author need to determine pepsin activity?
Answer: Pepsin activity was measured to investigate the potency of this enzyme toward meat protein with subsequent its effect on the texture of samples.
- For the rosemary oil compounds, please increase how’s each compounds (only main compounds or crucial is enough) affect to the fish quality.
Answer: The main components of rosemary EO used in the present study were eucalyptol, camphor, isoborneol, and α-pinene which possess great antimicrobial activity with a subsequent decrease in microbial population and TVBN. Similarly, Senanayake [51] found that eucalyptol, camphor, α-pinene, and bornyl acetate are the major compounds responsible for the antimicrobial activity of rosemary a matter which supports our findings (Table 3). Such clarification has been already mentioned in the discussion section (lines 428-433).
- The discussion of each parameter should be added. Now, it seem like author only report the results. Why protein content of fish increase after treated with additives? Please add the discussion.
Answer: The discussion explains why the protein content of fish increased after treatment with additives was added (lines 366-369). An explanation regarding the possible reason behind the high moisture content in the control basa was mentioned in the revised manuscript (lines 353-356). An explanation regarding the decrease in the pH value of the rosemary-treated sample was mentioned in the revised manuscript (lines 409-412). The significant decrease in TVBN values of rosemary-treated mackerel fillets was explained by Karoui and Hassoun [48] (It was mentioned in lines 417-418). An explanation regarding the significant decrease in TVBN of all used additives in the present study was mentioned in the revised manuscript (lines 426-430). The obtained low TBA value in all samples treated with the used additives was explained and clarified in the revised manuscript (lines 449-455). Moreover, rosemary EOs could maintain the color of fish fillets, due to the antioxidant activity of these EOs, which retard lipid oxidation and metmyoglobin formation [48]. This sentence was mentioned in the revised manuscript (lines 489-491). Explanation by Nurhayati et al. [29] was mentioned in the revised manuscript (lines 494-497). An explanation regarding the improvement of meat tenderness by the used additives was mentioned in the revised manuscript (lines 501-503). The explanation regarding the effect of used additives on cooking loss % was mentioned in the revised manuscript (lines 517-519 and 521-524). The explanation regarding the effect of used additives on microbial counts was mentioned in the revised manuscript (lines 622-627).
.
- Did proximate composition measure only on day0? Why? How to prepare the samples? Treat with the fish and then measure? If like this, why is highly impact to change the composition?
Answer: The proximate compositions of control and treated basa fish fillet were measured at day 0 of treatment as well as during storage however, the results indicated a non-significant NEGLECTED change during storage. It has a high impact on changing the composition due to the direct action of acidic pH of used additives on protein making denaturation with subsequent loss of water.
- Why author did not measure the proximate compositions during storage?
Answer: Special thanks for your comment. We measured the proximate compositions of control and treated basa fish fillet at day 0 as well as during storage however, the results indicated a non-significant NEGLECTED change during storage.
- The results were a little bit surprise, for example, SD. appeared at the low and equal level (Table4, all S.D. are only 0.01-0.03?)
Answer: Stander error (SE) in this table is ranged from 0.01-0.09 (0.01- 0.02- 0.03- 0.04- 0.05- 0.06- 0.09) due to nearly similar 9 readings (3 readings for each sample × 3 repetition at different times) for each trail.
- The units of each parameters did not appear.
Answer The units of each parameter have been already mentioned below table of protein deterioration and fat oxidation parameters.
- The citation with the standard did not provide likes how much perception of TVB-N in fish fillet? How much microorganisms that confirmed the fish is still safe, etc.- Why author did not measure the pathogen as per standard required since this experiment aimed to evaluate the shelf-life of the product.
Answer: Thank you so much for your valuable comments. The new title for the shelf life of basa fish fillet with new discussion for this title was added in the discussion section. In addition, the acceptable limits for TVBN, TBA, and bacterial counts according to the Egyptian Organization for Standardization and Quality Control were added (lines 675-684 and 688-693) as well as the reference added in the text as well as in the reference list.
Thank you again!

Reviewer 2 Report
Comments and Suggestions for Authors
Review of the manuscript foods-2659284, titled “Improvement of the microbial quality, lipid stability, fatty acid profile, organoleptic characteristics, and shelf life of basa (Pangasius bocourti) fillets during chilling storage using pepsin enzyme, rosemary, and citric acid”
Dear Authors,
The manuscript concerns very important issue of shelf-life of fish fillets. There were widely studied parameters of storage fillets treated by different compounds. There is high potential, however there were some weaker points of the work, which are listed below.
Abstract
In my opinion, the information about pepsin activity is not necessary in the abstract, because it is not the point of the work. Instead, some specific information about results should be added, how many days the shelf-life of fillets was prolonged, what composition of brines gave the best results.
Introduction
There should be highlighted the gaps in the subject and novelty of the work. Why had the Authors chosen the studied options?
L 54 – There should be “lack”, not “lake”, I suppose.
L 65-67 – Please, rephrase the sentence.
L76-85 – Application of citric acid and pepsin is poorly described.
L87 and 91 – There is mentioned marination technology, however, there is no information about marination in the title and aim of the work. Marinated fillets are usually soaked for some time (e.g. few days) in solutions of organic acids (and salt), mainly acetic acid (vinegar), what affects strongly the properties of fish meat, including ripening. In the introduction is lack of information of other methods of marination, if you would like to compare them…
Materials and methods
L95 – “occasions” – did you mean “batches”?
L103 – Characteristics of pepsin preparates should be given – kind, form, activity, protein content, specific activity. Usually commercial preparates are precisely characterized. If it was a solution of enzyme, there should be given solvent, concentration of enzyme and pH.
L108-116 – Preparation of fillets should be described better.
How were the solutions in which the fillets were immersed (brines) prepared? Tap or distilled water? Proportion of solutions to fillets? What was the immersion time? Were the fillets stored in these solutions or dried?
L119-126 – What was the hemoglobin solution pH? Was it optimal for pepsin activity?
L122 – Should be “mililiter” not “milimeter”.
L128 – Lack of brackets in the equation (according to this form only Ab is multiplied by 1000)
L136, 137 - Lack of space 5oC, and L98, 106, 114, 136, 140 and other – with space 50 oC. Please, unify the form throughout the manuscript.
L147 – Please, add numbers of procedures in the section or in the bibliography.
L148 – What did you mean by “however” in this sentence?
L164 – Fish extract, not filtrate.
L188 – Was it oil or chloroform extract, containing oil? What volume of the extract and/or what the mass of titrated oil was titrated?
L92 – x and X – it should be unified throughout the manuscript.
L208 – Which parts of fillets were used in the study? Were three repetitions enough for description of color parameters of different surfaces?
L205-207 – How many repetitions? Which parts of fillets were studied?
L209-211 – What conditions of cooking were used? What kind of cooking? Water, steam, oven, microwaving, etc.? The information about oven is given in description of sensory evaluation, but it should be added when cooking is mentioned for the first time. Temperature, time? Moreover, in my opinion, determinations made on cooked fillets should be clearly indicated.
L230-232 – Please, verify the validity of the use of brackets, they are not necessary in most of cases.
Results and discussion
L261-282 -Why have the authors examined pepsin activity and what impact did it have on the results? Was it determined only once? If not – were is standard deviation? What is the aim of comparing your commercial enzyme preparates with the activity of enzymes extracted from fish stomach? In my opinion, it is unjustified. Unless the activity was not provided by the manufacturer, then it is worth determining it. However, it is necessary to take into account the pH of the activity determinations and ipH of the tested samples. The impact of pepsin on meat properties should be included in the discussion.
Results shouldn’t be fully repeated in table and in text.
L277 – 280 – “pepsin activity seems to be temperature-dependent” - activity of all enzymes is temperature-dependent, it is usually precisely defined (after determination), especially in commercial enzymes. A certain temperature – which exactly? However, there is no information about optimum pH.
L286 – “concentration of each component varied markedly” – there is no concentration of component, only percentage of peak area in total peaks areas.
In table 2 – “Area sum %” – is it correct?
L305 – “The result” - shouldn’t be plural?
L311-320 – Please, take into account the influence of freezing; your raw material was frozen and thawed so it differ from fresh fillet.
Table 3 – please, verify the units – g% is old unit, it should be g/100 g
L311-L333 – In your research there is increase of moisture in fillets, so the results couldn’t be “in harmony” with results, were was reduction of moisture.
There is no discussion about reasons of protein content and other compounds.
L330-331 – The Authors written “in contrast to our findings (…) rosemary extract has no substantial effect”, however in your research results did no differ significantly (e.g. 85.03c and 85.19c). Please, verify your findings and the literature data.
Moreover, effect of combinations has not been discussed. It would be good, if you discussed for example why pepsin increased moisture and proteins etc.
What was the effect of pH on the activity of pepsin? Were the non-protein nitrogen levels determined?
L361-369 – Is it effect of low pH of pepsin? This is a protein, its activity is the highest in pH 2.5, but it’s not pH of pepsin itself. So the information about pH of pepsin solution is very important.
L409-410 – According to your results, pH values of these samples were about 6, are you sure that so high pH prevents microbial growth?
In marinated fish, when higher concentrations of citric acid and acetic acid are applied, pH is usually about 4, so there is low pH, in contrast to your samples.
L438-440 – Where are results of whiteness determination? Please, add results and description of determination/calculation method.
L451 – Please, remove name o authors and year, numbers of cited literature are enough.
L498 - ꞷ-9 and L533-542 n-3, n-6 – please unify to n-…
548-549 - Please, remove name o authors and year, number of cited literature is enough.
Conclusions
Please, add some specific information about shelf-life of fillets (how many days it was prolonged) and what composition of brines gave the best results.
Author Response
Reviewer≠2
Review of the manuscript foods-2659284, titled “Improvement of the microbial quality, lipid stability, fatty acid profile, organoleptic characteristics, and shelf life of basa (Pangasius bocourti) fillets during chilling storage using pepsin enzyme, rosemary, and citric acid”
Dear Authors,
The manuscript concerns very important issue of shelf-life of fish fillets. There were widely studied parameters of storage fillets treated by different compounds. There is high potential, however there were some weaker points of the work, which are listed below.
Dear Reviewer,
Special thanks for your efforts in reviewing our manuscript and your valuable comments which have greatly contributed to increasing its quality. We are delighted to read your words.
Please read below our answers to the raised concerns.
Abstract
In my opinion, the information about pepsin activity is not necessary in the abstract, because it is not the point of the work. Instead, some specific information about results should be added, how many days the shelf-life of fillets was prolonged, what composition of brines gave the best results.
Answer: Thank you so much for your valuable comment and I completely agree with you. The information about pepsin activity was deleted from the abstract section. The composition of brines that gave the best results was mentioned in lines 35-36. Based on the obtained results, the expected shelf life of control chilled basa fish fillet samples was 10 days because they deteriorated at the end of chilling storage (15 days). On the other hand, all treated basa fish fillet samples could maintain their sensorial attributes along with their low microbial load without any obvious deteriorative changes by the end of 15 days. This clarification was added in the abstract section as well as it has been already mentioned in the discussion section (lines 37-40).
Introduction
There should be highlighted the gaps in the subject and novelty of the work. Why had the Authors chosen the studied options?
Answer: Many thanks for your comment. This point has been already mentioned in the last paragraph of the introduction section (lines 110-112). Moreover, a new sentence of novelty point was added in the introduction section (lines 112-114).
L 54 – There should be “lack”, not “lake”, I suppose.
Answer: Thank you for your notice. The word was corrected.
L 65-67 – Please, rephrase the sentence.
Answer: rephrased (lines 75-77)
L76-85 – Application of citric acid and pepsin is poorly described.
Answer: A new paragraph for the application of citric acid and pepsin enzyme was added in the introduction section (lines 86-95). In addition, their references were added to the text and reference list.
L87 and 91 – There is mentioned marination technology, however, there is no information about marination in the title and aim of the work. Marinated fillets are usually soaked for some time (e.g. few days) in solutions of organic acids (and salt), mainly acetic acid (vinegar), what affects strongly the properties of fish meat, including ripening. In the introduction is lack of information of other methods of marination, if you would like to compare them…
Answer: A new paragraph about the marination process and different marination methods was added in the introduction section (lines 102-110).
Materials and methods
L95 – “occasions” – did you mean “batches”?
Answer: Yes
L103 – Characteristics of pepsin preparates should be given – kind, form, activity, protein content, specific activity. Usually commercial preparates are precisely characterized. If it was a solution of enzyme, there should be given solvent, concentration of enzyme and pH.
Answer: Bovine pepsin enzyme in the form of powder was used in the current study. The solvent was distilled water. The activity, protein content, and specific activity were measured, and the data were given in Table 2 because it was not provided by the manufacturer.
L108-116 – Preparation of fillets should be described better.
How were the solutions in which the fillets were immersed (brines) prepared? Tap or distilled water? Proportion of solutions to fillets? What was the immersion time? Were the fillets stored in these solutions or dried?
Answer: We used distilled water. The proportion of solutions was per Kg fish fillet (0.1% pepsin enzyme powder/Kg; 0.5% citric acid powder/Kg; 0.5% rosemary/Kg). All samples were stored in their solutions at 4 °C for 15 days. The required information was added in the revised manuscript (lines 131-133 and 144).
L119-126 – What was the hemoglobin solution pH? Was it optimal for pepsin activity?
Answer: 2% Hemoglobin in the buffer at pH =2 which is optimum for pepsin enzyme activity.
L122 – Should be “mililiter” not “milimeter”.
Answer: Corrected
L128 – Lack of brackets in the equation (according to this form only Ab is multiplied by 1000)
Answer: The brackets were added
L136, 137 - Lack of space 5oC, and L98, 106, 114, 136, 140 and other – with space 50 C. Please, unify the form throughout the manuscript.
Answer: The form was unified throughout the manuscript.
L147 – Please, add numbers of procedures in the section or in the bibliography.
Answer: Numbering was added to the procedures.
L148 – What did you mean by “however” in this sentence?.
Answer: Thank you so much for your comment. The word however was added only to decrease self-plagiarism in the materials and methods section.
L164 – Fish extract, not filtrate.
Answer: It is fish filtrate as a consequence of the filtration step. Since the fish homogenate was centrifuged firstly at 3000 rpm for 5 min, then the supernatant liquid was filtered through Whatman No. 1 filter paper.
L188 – Was it oil or chloroform extract, containing oil? What volume of the extract and/or what the mass of titrated oil was titrated?
Answer: It is oil. The weight of the extracted oil is not fixed. We can take for example 3 mL and then measure the amount of sodium hydroxide (mL) used for titration till obtain the pink color (end point). Put these values (W: Weight of oil (g) and mL of sodium hydroxide used for titration) in the calculation to obtain the Acid value.
L92 – x and X – it should be unified throughout the manuscript.
Answer: Unified
L208 – Which parts of fillets were used in the study? Were three repetitions enough for description of color parameters of different surfaces?
Answer: This is a mean value of color parameters for 3 readings from 5 different parts of fish fillet with 3 repetitions at different times (3 readings x 5 parts x 3 repetitions = 45 readings). The readings between the 5 different parts were non-significantly different.
L205-207 – How many repetitions? Which parts of fillets were studied?
Answer: 3 repetitions at different times and 6 readings for each repetition (3 x 6 = 18 readings). All samples were taken from the part below the dorsal portion of the fish fillet (upper middle) to be more reliable.
L209-211 – What conditions of cooking were used? What kind of cooking? Water, steam, oven, microwaving, etc.? The information about oven is given in description of sensory evaluation, but it should be added when cooking is mentioned for the first time. Temperature, time? Moreover, in my opinion, determinations made on cooked fillets should be clearly indicated.
Answer: All the required information was added in the revised manuscript (lines 244-245). The cooking loss percentage formula (the only determination made on cooked filets) was included in the manuscript.
L230-232 – Please, verify the validity of the use of brackets, they are not necessary in most of cases.
Answer: The unnecessary brackets were deleted.
Results and discussion
L261-282 -Why have the authors examined pepsin activity and what impact did it have on the results? Was it determined only once? If not – were is standard deviation? What is the aim of comparing your commercial enzyme preparates with the activity of enzymes extracted from fish stomach? In my opinion, it is unjustified. Unless the activity was not provided by the manufacturer, then it is worth determining it. However, it is necessary to take into account the pH of the activity determinations and ipH of the tested samples. The impact of pepsin on meat properties should be included in the discussion.
Answer: This value of pepsin enzyme activity was calculated from 3 measurements and a standard error was added to the table. We measured pepsin enzyme activity because it was not provided by the manufacturer. The optimum pH for the pepsin enzyme is 2. The impact of pepsin on meat properties was included in the discussion. The are lack of studies that mention the enzyme activity of commercial pepsin therefore, we compare the obtained results with those extracted from fish stomachs.
Results shouldn’t be fully repeated in table and in text.
Answer: Thank you so much for your valuable comment. Yes, I completely agree with you that it should not repeat the results in the text and table. Starting from table 4 till table 8 there is no repetition of the results. However, in Tables 2 and 3, I tried to just highlight in the text the significance of the obtained results to be clear.
L277 – 280 – “pepsin activity seems to be temperature-dependent” - activity of all enzymes is temperature-dependent, it is usually precisely defined (after determination), especially in commercial enzymes. A certain temperature – which exactly? However, there is no information about optimum pH.
Answer: The optimum temperature for the pepsin enzyme is 25 °C and the optimum pH is 2.
L286 – “concentration of each component varied markedly” – there is no concentration of component, only percentage of peak area in total peaks areas.
Answer: Thank you for your revision. Corrected into relative percentage.
In table 2 – “Area sum %” – is it correct?
Answer: Yes, it is correct
L305 – “The result” - shouldn’t be plural?
Answer: Corrected
L311-320 – Please, take into account the influence of freezing; your raw material was frozen and thawed so it differ from fresh fillet.
Answer: Thank you so much. Yes, I completely agree with you.
Table 3 – please, verify the units – g% is old unit, it should be g/100 g
Answer: Changed
L311-L333 – In your research there is increase of moisture in fillets, so the results couldn’t be “in harmony” with results, were was reduction of moisture.
Answer: Thank you so much for your notice. The sentence was corrected.
There is no discussion about reasons of protein content and other compounds.
Answer: The discussion explains the increase of moisture and protein content with a decrease in fat content added in the discussion section (lines 366-369)
L330-331 – The Authors written “in contrast to our findings (…) rosemary extract has no substantial effect”, however in your research results did no differ significantly (e.g. 85.03 and 85.19). Please, verify your findings and the literature data.
Answer: Special thanks for your revision. “In contrast to” was changed to “similarly”.
Moreover, effect of combinations has not been discussed. It would be good, if you discussed for example why pepsin increased moisture and proteins etc.
Answer: The discussion explains why pepsin increased moisture and protein content was added in the discussion section (lines 366-369).
What was the effect of pH on the activity of pepsin? Were the non-protein nitrogen levels determined? L361-369 – Is it effect of low pH of pepsin? This is a protein, its activity is the highest in pH 2.5, but it’s not pH of pepsin itself. So the information about pH of pepsin solution is very important.
Answer: The study the effect of pH on the pepsin activity was not numbered within the aim of the present research. Yes, it is the effect of low pH. This sentence (The decrease in pH value is due to the pepsin used having a pH of ±2.5) was mentioned exactly in the study conducted by Nurhayati et al. (2022).
Reference: Nurhayati, T.; Trilaksani, W.; Ramadhan, W.; Ichsani, S.P. The role of pepsin in improving the quality of surimi of red tilapia (Orechromis niloticus). Curr. Res. Nutr. Food Sci. 2022, 10, 584-594.
L409-410 – According to your results, pH values of these samples were about 6, are you sure that so high pH prevents microbial growth?
Answer: Special thanks for your comment. I mean herein the direct action of the acid pH of pepsin enzyme solution on the microbial growth of fish fillet samples.
In marinated fish, when higher concentrations of citric acid and acetic acid are applied, pH is usually about 4, so there is low pH, in contrast to your samples.
Answer: The pH value of citric acid-treated samples in this study was 5.83 at 0-time of examination and reached 6.34 at the end of chilling storage. These values are due to the lower citric acid concentration (0.5%) used in the present study than the concentration of citric acid (2.5% and 5%) used in the previous study (Klinmalai et al., 2021), which recorded pH values of citric acid-treated fish fillet samples ranged from 4.36–5.56.
Reference:
Klinmalai, P.; Fong-in, S.; Phongthai, S.; Klunklin, W. Improving the quality of frozen fillets of semi-dried gourami fish (Trichogaster pectoralis) by using sorbitol and citric acid. Foods 2021,
L438-440 – Where are results of whiteness determination? Please, add results and description of determination/calculation method.
Answer: Thank you so much for your revision. The whiteness index (WI) values were added in table 6 as well as the calculation method was added in the materials and methods section (lines 235-2380). Accordingly, the numbers of other equations were changed due to the addition of a new equation for whiteness index.
L451 – Please, remove name o authors and year, numbers of cited literature are enough.
Answer: removed
L498 -ꞷ-9 and L533-542 n-3, n-6 – please unify to n-…
Answer: changed
548-549 - Please, remove name o authors and year, number of cited literature is enough.
Answer: removed
Conclusions
Please, add some specific information about shelf-life of fillets (how many days it was prolonged) and what composition of brines gave the best results.
Answer: the required information was added in the conclusion section (lines 697-702).
Thank you again!

Reviewer 3 Report
Comments and Suggestions for Authors
The aim of this manuscript is to investigate the possibility of improving the microbial quality, lipid stability, fatty acid profile, organoleptic properties, and shelf life of basa fillets during storage using pepsin enzyme, rosemary, and citric acid.
As basa fillets have recently become more popular due to their low price and sensory properties, research that can contribute to a longer shelf life and thus better consumer acceptance is welcome.
The manuscript is clear, well structured, and overall well written. I cannot find fault with this manuscript.
Abstract: With the aim of increasing the scientific soundness of the results presented, I request that the abstract include numerical values for individual parameters studied as well as values of statistical significance. The sentence on pepsin enzyme activity (lines 32-33) is unnecessary and should be deleted.
Introduction: Please provide more references on the use of natural substances in the preservation of seafood. It is important to point out the novelty of this research compared to other research in this area.
The materials and methods section is well written and one might assume that the results are reproducible.
However, I recommend that a new table be introduced in this section to include labels and a description of the methods used (i.e. Table of Treatment Conditions). I recommend using the abbreviations introduced when displaying all results, referring to the table where they are explained. Consequently, the introduction of a new table requires a change in the numbering of all other tables and the display of all other results taking into account the new abbreviations.
Results and discussion section: I request that the display of all results be revised in line with my previous comment on the need to introduce a table of treatment designations.
When discussing the lipid quality indices, please refer to the recommended values for each parameter and compare the values obtained with other studies. Ln 552-553 please provide an explanation.
The conclusion is consistent with the evidence and arguments presented, but please include recommendations for future research and, if possible, a brief reference to the industrial relevance of the proposed research.
Comments on the Quality of English LanguageMinor editing of English language required
Author Response
Reviewer≠3
The aim of this manuscript is to investigate the possibility of improving the microbial quality, lipid stability, fatty acid profile, organoleptic properties, and shelf life of basa fillets during storage using pepsin enzyme, rosemary, and citric acid. As basa fillets have recently become more popular due to their low price and sensory properties, research that can contribute to a longer shelf life and thus better consumer acceptance is welcome. The manuscript is clear, well structured, and overall well written. I cannot find fault with this manuscript.
Answer:
Dear Reviewer,
Special thanks for your efforts in reviewing our manuscript and your valuable comments which have greatly contributed to increasing its quality. We are delighted to read your words!
Please read below our answers to the raised concerns.
Abstract: With the aim of increasing the scientific soundness of the results presented, I request that the abstract include numerical values for individual parameters studied as well as values of statistical significance. The sentence on pepsin enzyme activity (lines 32-33) is unnecessary and should be deleted.
Answer: The sentence on pepsin enzyme activity was deleted. Moreover, a new sentence was added in the abstract section (lines 37-40). Also, according to the reviewer's requirement, numerical values for the studied individual parameters were added in the new version of the abstract (lines 36-37).
Introduction: Please provide more references on the use of natural substances in the preservation of seafood. It is important to point out the novelty of this research compared to other research in this area.
Answer: The novelty point has been already mentioned in the last paragraph in the introduction section (lines 110-112). Moreover, a new sentence of novelty point was added in the introduction section (lines 112-114).
The materials and methods section is well written and one might assume that the results are reproducible. Thank you very much for your positive comments and appreciation!
However, I recommend that a new table be introduced in this section to include labels and a description of the methods used (i.e. Table of Treatment Conditions). I recommend using the abbreviations introduced when displaying all results, referring to the table where they are explained. Consequently, the introduction of a new table requires a change in the numbering of all other tables and the display of all other results taking into account the new abbreviations.
Answer: A new table for treatment conditions was added upon your suggestion as well a new sentence pointing to this Table was added in the material and methods section (line 135). Accordingly, the numberings of all other tables were changed. The abbreviations of all treatments were also added inside all tables. In addition, all abbreviations for treatments have been already mentioned in the abstract section (lines 29-30) as well as in the material and methods section (in treatment application, lines 138-142)
Results and discussion section: I request that the display of all results be revised in line with my previous comment on the need to introduce a table of treatment designations.
Answer: The display of all results was revised in line with the addition of a new table of treatment designations.
When discussing the lipid quality indices, please refer to the recommended values for each parameter and compare the values obtained with other studies.
Answer: The recommended values for Lipid Nutritional Quality Indices (LNQI) and comparing the values obtained with other studies for control samples have been already mentioned in the discussion section (lines 561-584). To the best of our knowledge, this study is the first concerning the effect of pepsin, rosemary EOs, or citric acid on LNQI of basa fillet which is considered a novel point for our study. In this regard, we cannot compare the obtained results regarding the effect of these additives on LNQI with the previous studies.
Ln 552-553 please provide an explanation.
Answer: There was an inverse relation between AI and TI values with the amount of PUSFA. Chakma et al. [44] observed that wild pangasius had low AI and TI values when compared with the farmed pangasius owing to the presence of PUFAs which were detected only in wild pangasius mentioned in the discussion section (line 536).
Thank you again!

Round 2
Reviewer 1 Report
Comments and Suggestions for Authors
The author has improved as suggested before
Author Response
Dear Reviewer,
Thank you again for your efforts and time in reviewing our manuscript and your valuable comments which have greatly contributed to increasing its quality!
Reviewer 2 Report
Comments and Suggestions for Authors
Review 2 of the manuscript foods-2659284
Dear Authors,
Thank you for your replies to my comments. Generally, you have taken into account most of my comments, but there is still some information missing in the revised text.
L128: Pepsin enzyme powder – should be added origin (“bovine”), Sigma-Aldrich offers more than one preparatus.
L128-133, L136-144: There is still lack of information about amount of solutions, proportion of solutions to fillets etc. Moreover, information is diverent: in lines 128-133 final concentration of compounds in relation to fillets mass (0.5%/Kg fillets) is given, and in lines 136-144 just concentration of solution is given. How it was exactly? Moreover, what exactly do you mean by “%/Kg”? What mass of fillets was used in each bag and what amount of solution was used in each bag? The amount of solution in relation to fillets mass is important (or just proportion), because it means different available amount of compounds.
It is necessary to provide information enabling the experiment to be repeated in another laboratory.
Moreover, there are different names/numbers of groups in the text L136-141, where the control sample is last mentioned and in the Table 1, where C is at the beginning.
“The first, second, and third groups were treated with 0.1% pepsin enzyme (E), 0.5% rosemary EOs (R), and 0.5% citric acid (CA), respectively. Meanwhile, the 4th group was treated with 0.1% pepsin enzyme + 0.5% rosemary EOs (E+R), the 5th group was treated with 0.1% pepsin enzyme + 0.5% citric acid (E+CA), and the 6th group was treated with 0.1% pepsin enzyme + 0.5% rosemary EOs + 0.5% citric acid (E +R +CA).” However, in the table there are Group 2, 3, 4, 5, 6 and 7, respectively. It should be unified.
L153 – pH should be added in the text
L179 – Contrary to the authors' response, no AOAC procedure numbers were added to item 18 (current number) in the bibliography.
L196 – I know that there was step of filtration, but after extraction – the main sense of this preparation was to deproteinate samples by addition of trichloroacetic acid (TCA) before further determination of TVBN. After centrifugation and filtration you obtained clear TCA-extract. Filtration was only the last step of extract preparation (mentioned in L193). So the resulted liquid is fish meat extract or TCA-extract, not only “fish filtrate” (in my opinion, this form describes much more types of extracts and samples without any other treatment then filtration, even brines).
Generally, could be accepted after minor revision.
Author Response
Reviewer≠2
Dear Authors,
Thank you for your replies to my comments. Generally, you have taken into account most of my comments, but there is still some information missing in the revised text.
Dear Reviewer,
Thank you for your efforts in reviewing our manuscript and your valuable comments which have greatly contributed to increasing its quality! Please read below our answers to the raised concerns.
Query 1: L128: Pepsin enzyme powder – should be added origin (“bovine”), Sigma-Aldrich offers more than one preparatus.
Answer 1: The word “bovine” was added in the revised manuscript in line 119.
Query 2: L128-133, L136-144: There is still lack of information about amount of solutions, proportion of solutions to fillets etc. Moreover, information is diverent: in lines 128-133 final concentration of compounds in relation to fillets mass (0.5%/Kg fillets) is given, and in lines 136-144 just concentration of solution is given. How it was exactly? Moreover, what exactly do you mean by “%/Kg”? What mass of fillets was used in each bag and what amount of solution was used in each bag? The amount of solution in relation to fillets mass is important (or just proportion), because it means different available amount of compounds.It is necessary to provide information enabling the experiment to be repeated in another laboratory.
Answer 2: The required information was added to the revised manuscript (lines 128-137).
Query 3: Moreover, there are different names/numbers of groups in the text L136-141, where the control sample is last mentioned and in the Table 1, where C is at the beginning. “The first, second, and third groups were treated with 0.1% pepsin enzyme (E), 0.5% rosemary EOs (R), and 0.5% citric acid (CA), respectively. Meanwhile, the 4th group was treated with 0.1% pepsin enzyme + 0.5% rosemary EOs (E+R), the 5th group was treated with 0.1% pepsin enzyme + 0.5% citric acid (E+CA), and the 6th group was treated with 0.1% pepsin enzyme + 0.5% rosemary EOs + 0.5% citric acid (E +R +CA).” However, in the table there are Group 2, 3, 4, 5, 6 and 7, respectively. It should be unified.
Answer 3: Thank you so much for your revision. The arrangement of groups was unified in the revised manuscript (lines 128-135) the same as in the Table.
Query 4: L153 – pH should be added in the text
Answer 4: pH was added in the revised manuscript line 151 (“in the buffer at pH of 2”).
Query 5: L179 – Contrary to the authors' response, no AOAC procedure numbers were added to item 18 (current number) in the bibliography.
Answer 5: We apologize for the misunderstanding. Moisture, protein, fat, and ash contents (g/100 g) of basa fish fillet samples were analyzed according to the official method of AOAC with procedure numbers AOAC 925.45, AOAC 981.10, AOAC 991.36, and AOAC 923.03, respectively [18]. AOAC procedure numbers were added to the text (lines 177-178) and reference list.
Query 6: L196 – I know that there was step of filtration, but after extraction– the main sense of this preparation was to deproteinate samples by addition of trichloroacetic acid (TCA) before further determination of TVBN. After centrifugation and filtration you obtained clear TCA-extract. Filtration was only the last step of extract preparation (mentioned in L193). So the resulted liquid is fish meat extract or TCA-extract, not only “fish filtrate” (in my opinion, this form describes much more types of extracts and samples without any other treatment then filtration, even brines).
Answer 6: The word filtrate was changed to extract in the revised manuscript line 195.
Generally, could be accepted after minor revision.
Thank you again for your valuable comments!